# Beyond Scalars: Concept-Based Alignment Analysis in Vision Transformers

**Johanna Vielhaben**[*]
Fraunhofer HHI

**Dilyara Bareeva**
Fraunhofer HHI

**Jim Berend**
Fraunhofer HHI

**Wojciech Samek**[†]
Fraunhofer HHI &
Technical University of Berlin

**Nils Strodthoff**[‡]
Carl von Ossietzky Universität Oldenburg

## Abstract

Measuring the alignment between representations lets us understand similarities between the feature spaces of different models, such as Vision Transformers trained under diverse paradigms. However, traditional measures for representational alignment yield only scalar values that obscure *how* these spaces agree in terms of learned features. To address this, we combine alignment analysis with concept discovery, allowing a fine-grained breakdown of alignment into individual concepts. This approach reveals both universal concepts across models and each representation's internal concept structure. We introduce a new definition of concepts as non-linear manifolds, hypothesizing they better capture the geometry of the feature space. A sanity check demonstrates the advantage of this manifold-based definition over linear baselines for concept-based alignment. Finally, our alignment analysis of four different ViTs shows that increased supervision tends to reduce semantic organization in learned representations.

## 1   Introduction

Vision Transformers (ViTs) [11] are gaining increased popularity as backbones for various computer vision tasks. There is a large zoo of pre-trained models covering different learning paradigms and supervision levels, with different capabilities [19] and thereby different internal representations. Consequently, understanding and comparing these representations is essential for practitioners when selecting pre-trained models, assessing robustness and generalization properties, and designing fine-tuning protocols. This opens up questions like: Where does the model representation change the most and how? Which concepts (i.e. dominant structures in representation space) are encoded in lower layers vs. upper layers? Does the model encode semantically similar concepts in spatial proximity to each other? How are the representations of model A aligned to that of model B across layers? One way to address these questions is by examining patterns in hidden activations and measuring *representational alignment*[35, 44], yet existing methods often provide only a single scalar to indicate similarity[41], leaving finer details unexamined.

In this paper, we propose a fine-grained alignment analysis based on concepts that structure the latent representation. This gives insights into universal concepts between representations of different layers or models, as well as how a single representation is structured. To achieve concept-based alignment we need solutions for concept discovery, and measuring the alignment between concept

---

[*]Now at Graphcore

[†]Corresponding author wojciech.samek@hhi.fraunhofer.de

[‡]Corresponding author nils.strodthoff@uol.de

39th Conference on Neural Information Processing Systems (NeurIPS 2025).

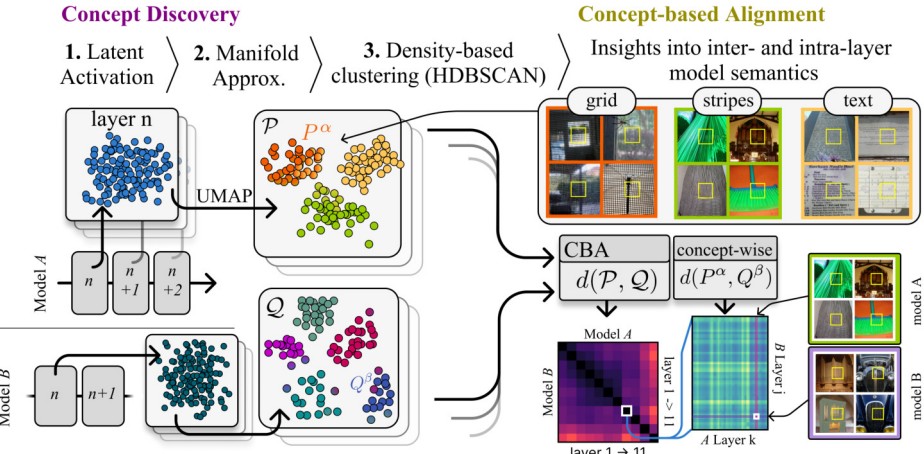

Figure 1: We combine concept discovery with alignment analysis for fine-grained insights into structures within and differences between latent activations. To this end, we investigate latent activations formed by intermediate layers, which according to the manifold hypothesis can be organized in terms of low-dimensional manifolds. We recover these concept manifolds using density-based clustering (HDBSCAN) applied to UMAP embeddings of the latent representations. The discovered structures in latent space allow to characterize a single layer and the formation of structures between layers.

proximity scores. Previous work on concept discovery has ranged from identifying individual neurons or other discrete units [1] to defining linear directions in feature space [14], with the most general approach so far viewing concepts as multi-dimensional *linear* subspaces [42]. However, there is growing evidence that linearity may be too strong an assumption [2, 8]. In contrast, we posit that to capture the underlying geometry of a representation more faithfully, concepts should be treated as the most general structure they can form—namely, *nonlinear manifolds*. For alignment measurement, previous methods typically characterize the similarity of similarities (e.g., via CKA [28]), collapsing the results into a single scalar score. We instead measure fine-grained distances between concept pairs. Specifically, we represent activations as proximity scores to discovered concepts and employ a generalized Rand index with pseudo-metric properties [24], which we partition into pairwise concept distances.

- We combine concept discovery with alignment analysis to reveal not only which concepts are universal between two representations but also how a single representation is internally structured (Sec. 2.2).

- We propose a novel definition of concepts as *nonlinear manifolds* and demonstrate—via a sanity check—the superiority of this definition for alignment analysis when compared to simpler and linear concept definitions (Sec. 4.2).

- In our concept-based alignment analysis of different ViTs, we find that their representations exhibit markedly different structures; specifically, increased supervision correlates with reduced semantic structure in the learned representations (Sec. 4.3).

## 2 Concept Discovery for Representational Alignment

This section consists of two parts. First, we introduce our novel non-linear concept definition – motivated by the manifold hypothesis – and describe our approach to discovering these concepts in latent activations (Fig. 1). Second, we show how this concept-based representation of hidden activations can be used to measure alignment across different models or layers, enabling a fine-grained analysis that goes beyond single-scalar metrics.

## 2.1 Concept Discovery

**Motivation** According to the manifold hypothesis, which is widely accepted in machine learning, many datasets, including image data that nominally lie in a high-dimensional space, can be described in terms of a few underlying latent factors and are thus concentrated on a (potentially disconnected) low-dimensional manifold embedded in high-dimensional space [20]. [32] shows how a neural network trained on a toy classification problem solves the task by transforming the topology of the input data, and layerwise reducing the Betti numbers of the class-wise components. We hypothesize that state-of-the-art vision models behave similarly and try to recover the connected components in the hidden representations, which we call *concepts*.

**Definition** We analyze the hidden representation at an intermediate feature layer of a neural network. To this end, we divide the model $f$ into two parts, $f = g_l \circ h_l$, where $h_l$ is the mapping to a hidden feature layer $l$. Our definition then relies on hidden representations $h_l(x_i) \in \mathbb{R}^{N' \times F}$ of input samples $x_i$ from a set $S$. $N'$ is the number of spatially separable elements in the representation, i.e., the number of tokens in a transformer model or the number of superpixels in a convolutional feature map. We spatially decompose the feature maps $h(x_i)$ into a set of $N = N' \cdot |S|$ feature vectors $\phi \in \mathbb{R}^F$. Previously, concepts have been mostly defined as linear structures [14, 46]. The most general linear structure would be affine subspaces, which would already represent an extension compared to the recently considered definition as linear subspaces [42]. In this work, we generalize this idea even one step further and define concepts as manifolds in the $F$-dimensional feature space.

**Definition 1** *We define a concept $C^\alpha$, as a manifold in $d$-dimensional feature space, represented by a point cloud $\{\boldsymbol{\phi}_j^\alpha\}$ consisting of the feature vectors $\boldsymbol{\phi}_j$ that lie on the concept manifold with index $\alpha$.*

**Benefits of concept manifold definition** In the following, we want to compute concept proximity scores by which we measure alignment. Incorrect assumptions about the structure of the concept manifold, e.g., assuming it has no curvature (affine subspaces) or it is spherical and the distance to the manifold can be estimated by the distance to the centroid, directly lead to distorted concept proximity scores and hence to distorted alignment. Later, in a sanity check our definition performs better than linear and non-linear spherical baselines for measuring representational alignment.

**Clustering for concept discovery** Having established our definition of concepts as manifolds in feature space, we now turn to the challenge of discovering these concepts through clustering. As stated above, we assume that feature vectors $\{\phi_i\}$ from a hidden representation are sampled from a set of low-dimensional concept manifolds $\{C^\alpha\}$. Recovering these concept manifolds in high-dimensional space ($F = 768$ in our experiments) is a challenging clustering problem. Therefore, we revert to density-based clustering on a low-dimensional embedding of the data [18, 22]. For this embedding, we utilize UMAP (Uniform Manifold Approximation and Projection) [29], a dimensionality reduction technique that preserves local and some global structure. Given that we have no a priori knowledge about the number of clusters, we employ HDBSCAN (Hierarchical Density-Based Spatial Clustering of Applications with Noise), which can handle clusters of varying densities [3]. HDBSCAN builds a hierarchy of clusters based on density, represented by a condensed tree, and allows for robust handling of noise, making it suitable for the possibly intricate structure of feature representation spaces. While UMAP does not fully preserve density, its ability to maintain the overall structure of the data makes it a valuable preprocessing step before applying HDBSCAN. We use the HDBSCAN implementation from [30].

**Concept proximity scores** We leverage soft clustering with HDBSCAN based on the condensed tree which is roughly a density function over the data points to compute fuzzy cluster membership as described in [30], which we formalize in the appendix for the reader's convenience. It is based on the distance to concept anchor points of each cluster and an outlier score, both derived from the condensed tree. We now have a fuzzy clustering $\mathcal{P}_{\{\phi\}} = \{\boldsymbol{P}(\boldsymbol{\phi}_0), \dots, \boldsymbol{P}(\boldsymbol{\phi}_N)\}$ with $n$ clusters, where $\boldsymbol{P} \in [0, 1]^n$ holds the concept proximity scores of each concept $C^\alpha$. We interpret the concept proximity scores $P^\alpha(\phi)$ as the probability that a feature vector $\phi$ belongs to a concept $P^\alpha$ in clustering $\mathcal{P}$. This approach contrasts with previous concept assignment paradigms [14, 42], which often rely on hard clustering, where each feature vector is assigned to a single concept, or linear methods that project onto specific concept directions, limiting the representation to a more rigid framework. In contrast, our soft clustering method allows for nuanced membership scores that reflect the degree

of belonging to multiple concepts. In the following, we refer to our concept discovery method as *NLMCD* (non-linear multi-dimensional concept discovery).

## 2.2 Concept-based Representational Alignment

We now address the question of measuring representational alignment based on the concept proximity scores derived from fuzzy clustering.

**Pseudo-metric between fuzzy clusterings** The concepts are at this point characterized by a probabilistic clustering $\mathcal{P}_{\{\phi\}} = \{\boldsymbol{P}(\boldsymbol{\phi}_0), \dots, \boldsymbol{P}(\boldsymbol{\phi}_n)\}$, where $\boldsymbol{P}(\boldsymbol{\phi}_i) = [P^1(\boldsymbol{\phi}_i), \dots, P^n(\boldsymbol{\phi}_i)]$. We want to measure the similarity between two probabilistic clusterings $\mathcal{P}, \mathcal{Q}$ from two different representations to evaluate how aligned their concepts are. For this purpose, we leverage an extension of the pair-based Rand index generalized to fuzzy clusterings proposed in [24]. The original Rand index counts the number of concordant pairs (either two points are paired or not paired both clusterings) and disconcordant pairs (two points are paired in one clustering but not in the other). The distance between probabilistic clustering $\mathcal{P}, \mathcal{Q}$ is based on a generalized degree of concordance that is based on the *distance between two membership vectors* $d_{ms}(\boldsymbol{P}(\boldsymbol{\phi}_i), \boldsymbol{P}(\boldsymbol{\phi}_j))$:

$$d_{cross}(\mathcal{P}, \mathcal{Q}) = \frac{2}{n(n-1)} \sum_{i<j} |d_{ms}(\boldsymbol{P}(\boldsymbol{\phi}_i), \boldsymbol{P}(\boldsymbol{\phi}_j)) - d_{ms}(\boldsymbol{Q}(\boldsymbol{\phi}_i), \boldsymbol{Q}(\boldsymbol{\phi}_j))| \tag{1}$$

A commonly used choice for the distance $d$ is $d_{ms}(\boldsymbol{P}(\boldsymbol{\phi}_i), \boldsymbol{P}(\boldsymbol{\phi}_j)) = 1 - ||\boldsymbol{P}(\boldsymbol{\phi}_i) - P(\boldsymbol{\phi}_j)||_1$ [9]. Finally, we refer to the similarity between two clusterings, derived from the uncovered concepts, as *Concept-Based Alignment* (CBA):

$$\text{CBA} = 1 - d_{cross}(\mathcal{P}, \mathcal{Q}) \tag{2}$$

We choose this measure because $d_{cross}(\mathcal{P}, \mathcal{Q})$ is a pseudo-metric satisfying desirable properties[4] that ease interpretation Also, when $\mathcal{P}, \mathcal{Q}$ are crisp partitions, CBA reduces to the original Rand index.

**Distance between single clusters** In contrast to conventional measures for representational alignment that yield a single scalar value, our approach provides a more nuanced measure of representational alignment by assessing differences between pairs of single clusters. To measure the distance between two clusters $P^\alpha, Q^\beta$ from two clusterings $\mathcal{P}, \mathcal{Q}$, we decompose the distance in Eq. 1 into the contribution of single concepts $P^\alpha, Q^\beta$ and measure the *pairwise similarity between the membership scores* of each feature

$$d_{cross}(P^\alpha, Q^\beta) = \frac{2}{n(n-1)} \sum_{i<j} ||P^\alpha(\boldsymbol{\phi}_i) - P^\alpha(\boldsymbol{\phi}_j)| - |Q^\beta(\boldsymbol{\phi}_i) - Q^\beta(\boldsymbol{\phi}_j)|| \tag{3}$$

Due to the absolute value in Eq. 1, summing over all pairs $\alpha, \beta$ does not yield the total $d_{cross}(\mathcal{P}, \mathcal{Q})$, but by the triangle inequality $\sum_{\alpha, \beta} d_{cross}(P^\alpha, Q^\beta) \geq d_{cross}(\mathcal{P}, \mathcal{Q})$ the sum is an upper bound for the overall distance between two clusterings.

## 3 Related work

**Alignment** Representational alignment measures are categorized, with a particular emphasis on Centered Kernel Alignment (CKA) in [28]. CKA evaluates the similarity of similarities, either linearly or under a non-linear kernel. Similarly, [10] measure alignment through the similarities of binary k-nearest neighbor adjacency matrices, which resembles CKA with a narrow Gaussian kernel. Our method relates to CKA in that it condenses these similarities into clusters and subsequently measures the similarity between these clusterings. In parallel to the initial release of this work, [26] introduced a related concept-based alignment measure which relies on class-wise one-dimensional linear concepts. In contrast, we analyze alignment across classes based on non-linear, multidimensional concepts.

**Concept discovery** Most existing methods model concepts as linear directions [17, 46, 15, 14, 23, 16]. Generalizing this definition, [42] suggest that concepts can be represented more faithfully as multidimensional linear subspaces.While the above approaches operate unsupervised, without access to concept labels, supervised concept discovery methods have explored more flexible geometries:

---

[4]1) Identity: $d(x, x) = 0$ for all $x$, 2) Symmetry: $d(x, y) = d(y, x)$ for all $x, y$, 3) Triangle Inequality: $d(x, z) \leq d(x, y) + d(y, z)$ for all $x, y, z$.

[39] estimate concept manifolds as multi-dimensional ellipsoids, [7] employ kernel classifiers for nonlinear concept discovery and report improvements over linear methods, and also [8] find evidence for the existence of non-linear features. Unlike these approaches, our main goal in concept discovery is representation summarization for alignment measurement, rather than interpretability or feature enumeration. For this reason, we employ the most general, non-linear concept definition.

**Comparison of Vision Models**  On the one hand, alignment measures such as CKA have been used to compare the representations of various vision models, including ViTs and ResNets trained on different tasks or datasets [6, 5]. This has been combined with the analysis of patterns in attention maps [44, 35, 33], visualization of feature maps [12, 38], and linear probes [38]. On the other hand, downstream performance is analyzed to guide the selection of pre-trained models for transfer learning [27, 19].

## 4  Results

We evaluate concept discovery in Sec. 4.1, check the superiority of our new concept definition over linear and simplified baselines for concept alignment analysis in Sec. 4.2, and perform a concept-alignment analysis between four ViTs in Sec. 4.3.

### 4.1  Concept discovery

**Experimental setup**  First, we outline the details of the concept discovery procedure as described in Sec. 2.1 based on UMAP embeddings and HDBSCAN clustering. For concept discovery and later analysis of representational alignment, we use a random subset of 25 % of the ImageNet train set, stratified samples across all 1000 classes. We study four different ViTs [11] with the same architecture (base, patch size 16, input size 224) but different training objectives and training datasets: **FS** [40] trained supervised on ImageNet-1k [37] classification, **CLIP** [34] trained to contrast images and text from WebImageText [34], **DINO** [4] trained on ImageNet-1k to enforce consistency between augmented views of the same image, and **MAE** [21] trained to reconstruct missing pixels of input data from ImageNet-1k. We perform concept discovery separately for the sequence (SEQ) and the CLS token. We extract activations at the last MLP layer of each of the twelve transformer blocks. Due to computational constraints of UMAP and HDBSCAN limiting the number of tokens for clustering, we reduce the 196 SEQ tokens per image to a single representative token, which also facilitates comparison with CLS experiments. We obtain this token by average pooling over a central $4 \times 4$ block of tokens, assuming the image center contains more diverse concepts, while peripheral regions may predominantly capture repetitive background elements. Further, for SEQ tokens, we discard the last block as for the considered models only the CLS token in the final layer enters the loss. Hyperparameter tuning for UMAP and HDBSCAN (based on DBCV) and the final settings are detailed in the appendix.

**Evaluation**  We assess concept discovery using four metrics. First, we report the rate of points classified as noise by HDBSCAN. Second, we compute the density-based cluster validity index (DBCV) [31], which compares intra- vs. inter-cluster density and ranges from $[-1, 1]$ (higher is better).[5] Third, we evaluate how well the UMAP-based clustering preserves distances by calculating the root mean squared error between the original and embedded distance matrices, normalized by the mean pairwise distance (NRMSE). Finally, we test the robustness of our approach by measuring the alignment between two runs – with fixed model weights and input samples, but different initializations for UMAP and HDBSCAN – using CBA from Eq. 2.

**Results**  Turning to the results on embedding and clustering quality in Fig.2, we find that NRMSE remains consistently low across most layers and models. Exceptions include the CLIP CLS representation in layer one and the DINO CLS representation between layers six and eleven, where NRMSE rises. The DBCV scores range between 0.4 and 0.7, indicative of medium clustering quality, yet remain fairly consistent across both models and token types (SEQ vs. CLS). Given the inherent complexity of the clustering task, this level of performance is reasonable, supported by the qualitative

---

[5]We do not weight the average DBCV by cluster size as proposed in Moulavi et al. [31] so that noise rate and DBCV remain independent.

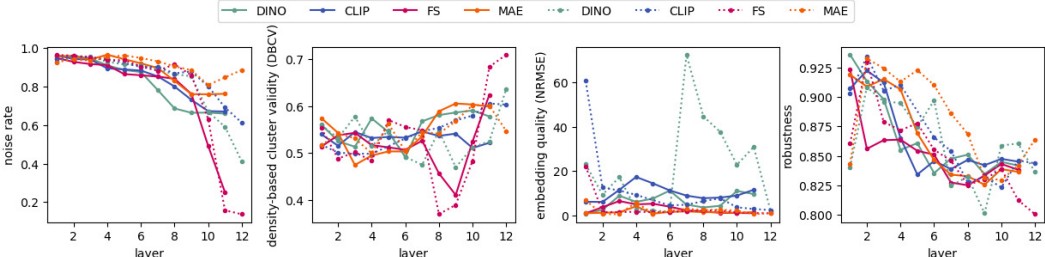

Figure 2: Quality of concept discovery: The noise rate is the ratio of points classified as noise in HDBSCAN. DBCV is a density-based clustering validity index that contrasts intra- vs inter-cluster density with scores in $[-1, 1]$ where higher is better. NRMSE measures the root mean squared error between the distance matrix of the original and embedded activations, normalized by the average distance in the original embedding, and shows how faithfully the UMAP embedding captures the geometry of the representation. Robustness is measured between two runs by concept-alignment from Eq. 2. Results are across layers for CLS (dotted) and SEQ (solid) token representations of the models introduced in Sec. 4.1.

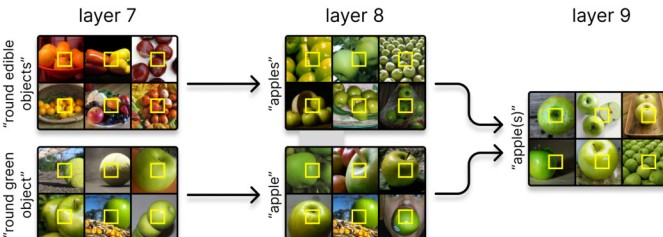

Figure 3: Concept formation graph for the concept "apple(s)" in layer 9 of the FS model. Each concept is represented by six randomly sampled images containing a token assigned to that concept (highlighted in a yellow frame).

results in Fig.3 and Fig. 4. The noise rate is relatively high overall but diminishes in deeper layers. Insufficiently dense sampling, constrained computationally by UMAP and HDBSCAN, may prevent some noisy regions from crystallizing into valid concept clusters. Robustness decreases for all models across layers but stagnates at around $0.84$ for most models in the late layers. Finally, two random baselines based on a randomly initialized model and/or randomized input (detailed in the appendix) exhibit weaker clustering validity, confirming that concept discovery relies substantially on both the learned representations and the underlying input distribution.

**Concept formation graphs** Finally, for a qualitative evaluation of our concept discovery method, we construct concept formation graphs (CFGs). These are unweighted, directed graphs that show how tokens transition from one concept to another across consecutive layers. In Fig. 3, we illustrate how the "apples" concept develops from layer seven to layer nine in the FS model. Additional examples for other models and the detailed algorithm for CFG construction are provided in the appendix.

### 4.2 Sanity checking concept structure for alignment and CKA comparison

**Setup** We use a sanity check to demonstrate how concept-based alignment analysis benefits from representing concepts as non-linear manifolds, comparing these results to alternative concept definitions and discovery methods. Additionally, we compare CBA from Eq. 2 against CKA as an established representational alignment measure that offers a single scalar to indicate alignment. This sanity check relies on the assumption by [25] that adjacent layers should align more strongly than distant ones. For each layer, we compute the weighted Kendall's Tau correlation [43] between alignment scores and layer distances, separately for upstream and downstream layers. We use hyperbolic weights to prioritize alignment between closer layers (whose alignment is more informative) and separate upstream from downstream layers to accommodate different rates of representational change (e.g., layer six may differ more from layer seven than from layer four). Averaging these correlations

Table 1: Sanity check for concept alignment, based on weighted Kendall Tau [43] between alignment and layer distance. We compare the suitability of NLMCD for Concept-Based Alignment (CBA) concepts against other methods: one-dimensional linear subspaces (PCA), multi-dimensional linear subspaces (MCD), and spherical non-linear concepts (KMeans). Additionally, we compare CBA against CKA. Results within the same standard error interval as the top score for each model are **bold** and those CBA results within the same interval as NLMCD-CBA are *italic*. NLMCD consistently outperforms other concept approaches. While NLMCD-CBA and CKA are en-par, CBA offers the advantage of fine-grained concept-based alignment.

| | SEQ | | | | CLS | | | |
| | FS | CLIP | DINO | MAE | FS | CLIP | DINO | MAE |
|---|---|---|---|---|---|---|---|---|
| PCA-CBA | 0.91(2) | 0.91(3) | *0.88(4)* | 0.84(3) | ***0.92(2)*** | 0.91(2) | 0.78(5) | 0.78(4) |
| MCD-CBA | 0.90(4) | 0.92(4) | 0.85(4) | 0.87(4) | 0.82(4) | 0.73(4) | 0.62(5) | 0.73(5) |
| KMeans-CBA | 0.94(2) | 0.82(4) | *0.87(4)* | ***0.98(1)*** | ***0.96(3)*** | 0.73(1) | ***0.89(5)*** | 0.86(4) |
| **NLMCD-CBA** | *0.97(1)* | *0.98(1)* | *0.92(2)* | *0.98(1)* | *0.93(2)* | *0.96(1)* | *0.91(2)* | *0.94(2)* |
| CKA | **0.98(1)** | 0.94(2) | **0.99(0)** | **0.99(1)** | **0.93(2)** | **0.97(1)** | **0.89(3)** | **0.93(3)** |

across layers verifies whether the alignment measure reflects the expected structural relationships. Using this sanity check, we compare CBA based on NLMCD concepts against one-dimensional linear subspaces discovered by PCA [46, 14], multi-dimensional linear subspaces discovered by MCD [42], and spherical non-linear concepts discovered by KMeans clustering [14]. For the linear subspaces, we project feature vectors onto the concept subspace and clip negative values to zero to obtain soft membership scores (arguing that a vector pointing in the opposite direction of a concept is inactive). For KMeans concepts, we measure concept proximity by the euclidean distance to the cluster centroid. We also normalize concept membership scores $P^{\alpha'} = P^{\alpha} / \sum_{\alpha} P^{\alpha}$ as their sum is required to be less bounded by one $\sum_{\alpha} P^{\alpha} \leq 1$ in Eq. 1. There is no direct way to estimate the number of concepts for PCA, MCD and KMeans, so we use all $F = 768$ components for PCA as a conservative baseline, and the number of concepts discovered by NLMCD for MCD and KMeans discovery.

**Results** We present the sanity check in Tab. 1 for SEQ and CLS token alignment. We find that for SEQ tokens, NLMCD shows higher scores than linear concepts except for DINO, where PCA can match its performance. Further, only for DINO and MAE simple nonlinear KMeans is en par with NLMCD. Similarly, for CLS tokens, NLMCD mostly outperforms other concept methods, while only for FS can PCA, and for CLIP and MAE can KMeans match its performance. To complement this quantitative evaluation of how well different concept definitions and discovery methods suit alignment measurement, we visualize PCA, MCD, and KMeans concepts and their geometric structure (based on our concept distance measure) in the appendix. By visual inspection, we find that the non-linear NLMCD and KMeans concepts are most coherent within each concept cluster and also show the highest coherence across groups of closely aligned concepts. When comparing to CKA, NLMCD-CBA achieves similar performance for both SEQ and CLS tokens. However, it offers the additional advantage of providing a fine-grained, concept-level view of alignment, unlike the single-scalar nature of CKA.

## 4.3 Concept Alignment Analysis

We now investigate concept-based alignment described in Sec. 2.2 between representations across layers and models. We structure the analysis into *intra-model* and *inter-model* to answer questions 1) - 4) openend in the introduction. Due to limited space, we focus on SEQ representation and defer the CLS representation analysis to the appendix. First, we analyze how representations are transformed within one model and how they are structured across layers.

**Intra-model: Where does the model representation change the most and how?** First, we focus on the intra-model alignment heatmaps between SEQ representations across layers measured by CBA from Eq. 2 in the upper row of Fig. 4. In Fig. 5, we complement these findings with additional concept clustering characteristics: alignment of concepts with ImageNet-1k labels, the intrinsic dimensionality of each concept, and the number of concepts per layer. In CLIP, DINO, and MAE, we observe a prominent break between layers one to six and layers six to eleven. The number of

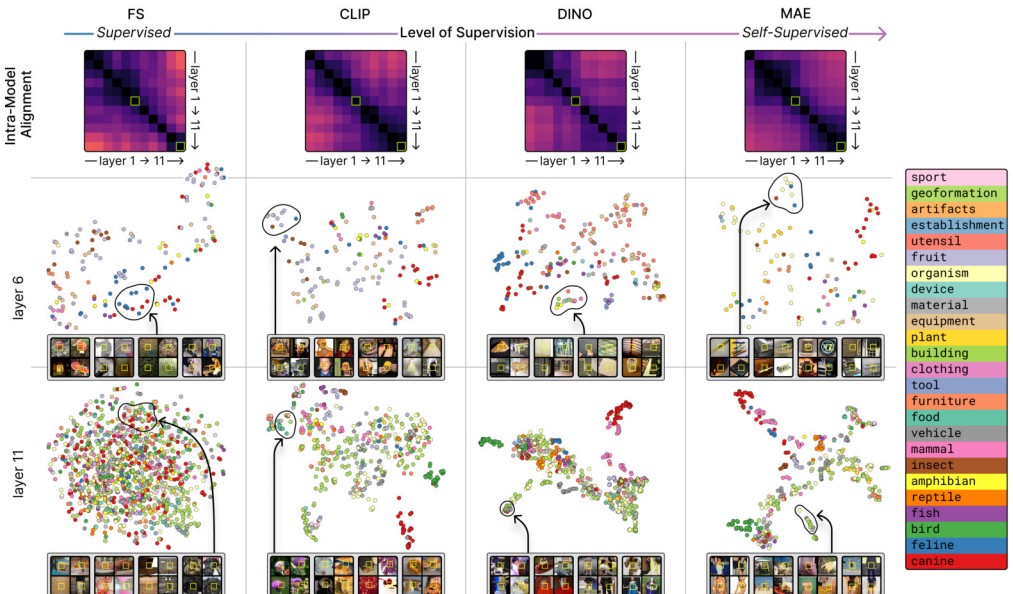

Figure 4: Intra-model relationships of SEQ representations. **Upper row:** We show CBA from Eq. 2 to visualize how representations are transformed across layers of the models introduced in Sec. 4.1 (darker pixels correspond to higher alignment). We observe a nucleation process between layer 9 and 10 in FS and smoother processing split into two major blocks between layer 1-6 and 6-11 in CLIP, DINO and FS. **Center and bottom row:** we zoom into the representations at layer 6 and 11 and partition the scalar CBA alignment into single concepts. We show a UMAP embedding constructed from the pairwise distance of concept measured by $d_{cross}(P^\alpha, P^\beta)$ from Eq. 3. Each point in this *concept atlas* corresponds to a distinct concept $P^\alpha$. To convey their meaning, for some concepts, we show four random input tokens from the members of the concept cluster $P^\alpha$ marked by a yellow box in the entire image. The stronger the supervision during ViT training ranging from FS, over CLIP to DINO and MAE, the less semantically organized are the representations at layer 11.

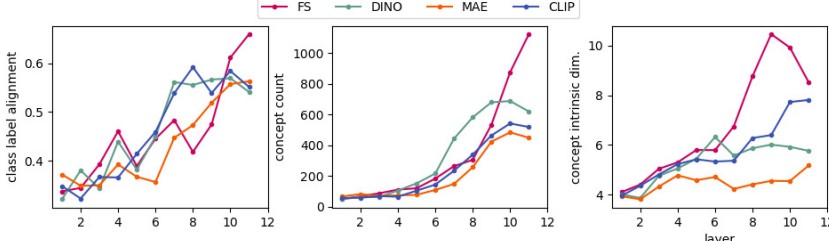

Figure 5: To supplement the intra-model alignment analysis, we evaluate alignment between concepts and ImageNet-1k class labels, (based on CBA from Eq. 2), concept count, and the average intrinsic dimensionality (based on [13]) across concepts.

concepts increases smoothly in the first part but rises sharply from layer seven onward. At that same point, class label alignment shows a marked increase, whereas the average concept dimensionality slightly decreases for DINO and MAE but increases for CLIP. In contrast, FS shows a distinct transformation between layers nine and ten, leading to a sudden jump in both class alignment and the number of concepts, accompanied by higher intrinsic dimensionality. This shift results in a lower alignment between representations in the last two layers and earlier layers and suggests the formation of distinct class-specific concepts, mirroring the "nucleation process" previously reported for ResNets by Doimo et al. [10].

**Intra-model: Which concepts are encoded in lower layers vs. upper layers? How structured are the representations?** We now zoom in on the representational alignment scores by examining pairwise distances among individual concepts. Specifically, we use a *concept atlas* – a two-dim.

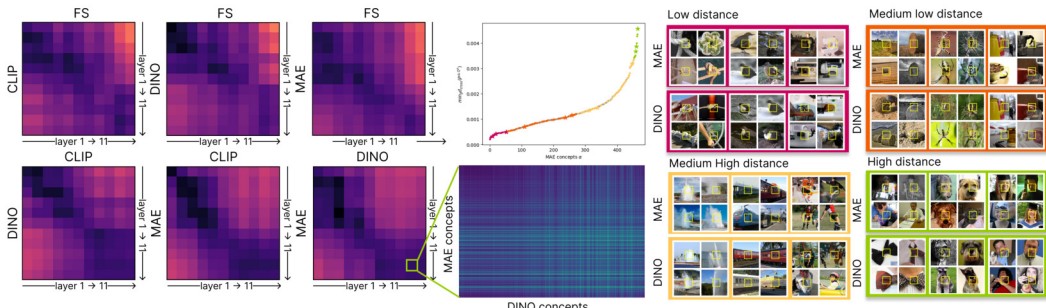

Figure 6: Inter-model relationships of SEQ representations. **Left:** We show CBA from Eq. 2 to visualize how representations differ between models(darker pixels correspond to higher alignment). **Right:** We zoom into the concept-wise distances $d_{cross}(P^\alpha, P^\beta)$ from Eq. 3 between the representation of layer 10 in MAE and DINO. Alongside the pairwise distance matrix $d_{cross}(P^\alpha, Q^\beta)$, we present randomly chosen examples of concepts from MAE and their nearest matches in DINO, and the distribution of the distances between nearest matches. We illustrate some random example pairs with low to high distance (marked by a star in the distance distribution plot). Overall, concepts that show a higher pairwise distance also appear slightly more distinct visually.

UMAP embedding of the pairwise concept distance matrix – to visualize the structures concepts form at layers six and eleven across of all models (Fig. 6). We color-code each concept cluster according to WordNet categories derived from ImageNet-1k labels [6] to guide the eye. At layer six, all ViT models display a structured set of concepts which mostly focus on lower-level structures, shapes, and object parts, rather than semantic WordNet-based groupings. In comparison, at layer eleven, CLIP, DINO, and MAE concepts show strong semantic organization, illustrated by clearly separated canine concepts and coherent groupings of human body parts such as neck, shoulder, and legs. In contrast, FS at layer eleven appears less semantically organized – fully supervised training may push similar concepts apart (for instance, different dog breeds) to reduce confusion and optimize for task-specific accuracy. This, however, may have negative implications for generalization to other tasks.

**Inter-model relations**     Second, we analyze how the representations between two different models differ and present $CBA$ from Eq. 2 between all layers of the models in the upper part of Fig. 6. We observe higher alignment between the self-supervised models DINO and MAE than with CLIP and the FS model in the alignment heatmaps. Further, layers of the first are more aligned than those of the second half across all models pairs. We conclude that basic foundational features are learned similarly across models, while later layers diverge as the models specialize to concepts serving their pre-training task.

**Inter-model: How is the representation of model A similar to that of model B?**     We now select a pair of layers from two different models (layer ten in MAE and DINO) for a closer look at concept-resolved differences in their representations. Alongside the pairwise distance matrix $d_{cross}(P^\alpha, Q^\beta)$, we present randomly chosen examples of concepts from MAE and their nearest matches in DINO, as well as the distribution of the distances between nearest matches. Overall, concepts that show a higher pairwise distance also appear slightly more distinct visually. Interestingly, around 62% of concepts from MAE all match to only three *singular* concepts in DINO, which are marked in gray in the distance distribution plot in Fig. 6 and excluded from example sampling. Upon inspection, those singularly matched concepts have no visual resemblance. Instead, the singular concepts in DINO exhibit low total activation across samples. Consequently, as our distance measures not only how concepts activate similarly, but also how they similarly do not activate, they are closest match to a variety of MAE concepts that have no close match otherwise – by virtue of their shared lack of activation. Distances to these singular concepts span a range, depending on the activation strength of the concept from the first model. We observe similar behavior for other pairs of layers.

---

[6]These are derived from the ImageNet-1k labels of the images from which the patches were extracted. We first map these labels to more abstract categories (provided in the appendix). Second, we perform a majority vote among all patches in a cluster to assign the category. This labeling is only a proxy and may not accurately reflect the actual content of the patches—e.g. a patch might show grass on which an animal stands.

# 5   Conclusion

We propose a novel approach that combines concept discovery with representational alignment analysis in ViTs. With concept-based alignment analysis, we answer the questions raised in the introduction and examine the structure of feature spaces of different ViTS, as well as fine details between the concepts of two different models. These insights are not available through traditional scalar alignment measures. Understanding the structured nature of latent spaces can guide practitioners in choosing models that not only perform well on benchmark datasets but also exhibit robust feature representations for downstream tasks. For instance, the nucleation process in FS emphasizes the importance of model structure over mere classification accuracy when selecting a pre-trained model.

**Limitations**   The computational scalability of HDBSCAN limits the sampling of feature vectors which makes undersampled concept regions appear as noise. The limited variability of ImageNet-1k might obfuscate the meaning of a concept, e.g. when a concept represents a color but there are only dog patches of that color. Further, the computational complexity of CBA requires subsampling of feature vectors.

**Acknowledgments**   This work was supported by the European Union's Horizon Europe research and innovation programme (EU Horizon Europe) as grants [ACHILLES (101189689), TEMA (101093003)]; and the German Research Foundation (DFG) through research unit DeSBi [KI-FOR 5363] (project ID: 459422098) and grant SELPHY-TS (project ID: 553038473).

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

# A HDBSCAN

After concept discovery with HDBSCAN, we compute concept proximity scores $\mathcal{P}_{\{\phi\}} = \{\boldsymbol{P}(\boldsymbol{\phi}_0), \ldots, \boldsymbol{P}(\boldsymbol{\phi}_N)\}, \boldsymbol{P} \in [0,1]^n$ holds the concept membership scores $P^\alpha(\boldsymbol{\phi})$ of each concept $C^\alpha$. These rely on the implementation of soft clustering with HDBSCAN from [30], which we formalize here for the reader's convenience.

**Clustering** HDBSCAN first transforms the feature space using a density-informed metric called *mutual reachability distance*

$$\text{MRD}(\boldsymbol{\phi}_i, \boldsymbol{\phi}_j) = \max(\text{coreDistance}_k(\boldsymbol{\phi}_i),$$
$$\text{coreDistance}_k(\boldsymbol{\phi}_j), d(\boldsymbol{\phi}_i, \boldsymbol{\phi}_j)) \tag{4}$$

where $\text{coreDistance}_k(\boldsymbol{\phi})$ is the distance between a point $\boldsymbol{\phi}$ and its $k$-nearest neighbor. Based on the mutual reachability distance between all pairs, a minimum spanning tree is constructed that connects all points and minimizes the sum of the edges weighted by MRD. From this, a hierarchical tree is constructed via robust single linkage clustering. The hierarchical tree is condensed by eliminating insignificant clusters and simplifying the hierarchy. This is achieved by selecting a range of *persistence* values $\lambda$, which are the inverses of the mutual reachability distances ($\lambda = 1/\text{MRD}$). Clusters that persist over significant ranges of $\lambda$, i.e. they are stable across multiple density levels, are retained, while clusters that exist only over narrow ranges of $\lambda$ are considered noise and pruned from the tree. The result is a condensed tree that focuses on the most significant clusters. Finally clusters are extracted from the condensed tree either based on their stability across different density levels or simply the leaf nodes are identified as clusters.

**Soft clustering with HDBSCAN** The soft cluster membership scores combine a distance-based membership with and an outlier score.

For the distance-based membership to cluster $C^\alpha$, first $k$ exemplar points $\{\boldsymbol{\phi}_i^\alpha\}$, $i \in [1, k]$, are extracted. A single centroid is not enough to characterize a cluster as its shape can be arbitrary. The exemplar points are the points within the leaf nodes beneath cluster $C^\alpha$ with maximum persistence $\lambda$ in the condensed tree, i.e. the densest points where the cluster persists. Then, the distance membership score between a point $\boldsymbol{\phi}$ and a cluster $C^\alpha$ is the inverse minimum distance across the exemplar points $\{\boldsymbol{\phi}_i^\alpha\}$,

$$M^\alpha(\boldsymbol{\phi})_{\text{dist}} = \frac{[\min_i(d(\boldsymbol{\phi}, \boldsymbol{\phi}_i^\alpha))]^{-1}}{\sum_\beta [\min_j(d(\boldsymbol{\phi}, \boldsymbol{\phi}_j^\beta))]^{-1}} , \tag{5}$$

normalized across all clusters. The outlier-based membership compares a point's membership persistence to the total persistence of a cluster:

$$M^\alpha(\boldsymbol{\phi})_{\text{membership}} = \frac{\lambda_{\boldsymbol{\phi} \to C^\alpha} - \lambda_{\text{birth}}^{C^\alpha}}{\lambda_{\text{max}}^{C^\alpha} - \lambda_{\text{birth}}^{C^\alpha}} . \tag{6}$$

Here, $\lambda_{\text{birth}}^{C^\alpha}$ is the persistence value at which cluster $C^\alpha$ first appears, i.e. its birth point in the condensed tree and $\lambda_{\boldsymbol{\phi} \to C^\alpha}$ is the persistence value at which point $\boldsymbol{\phi}$ would join cluster $C^\alpha$. Finally, distance and outlier-based membership are combined with stronger emphasis on outlier-based membership,

$$M^\alpha(\boldsymbol{\phi}) = (M^\alpha(\boldsymbol{\phi})_{\text{dist}})^{1/2} \cdot (M^\alpha(\boldsymbol{\phi})_{\text{membership}})^2 , \tag{7}$$

and normalized $M_{\text{norm}}^\alpha(\boldsymbol{\phi}) = M^\alpha(\boldsymbol{\phi})/\sum_\beta M^\beta(\boldsymbol{\phi})$. This membership score $M_{\text{norm}}^\alpha(\boldsymbol{\phi})$ can be interpreted as the probability that a point $\boldsymbol{\phi}$ belongs to cluster $C^\alpha$, given that the point belongs to some cluster,

$$M_{\text{norm}}^\alpha(\boldsymbol{\phi}) \equiv P(\boldsymbol{\phi} \in C^\alpha \mid \exists \beta : \boldsymbol{\phi} \in C^\beta) . \tag{8}$$

We want to compute the joint probability $P(\boldsymbol{\phi} \in C^\alpha)$, which includes the probability that $\boldsymbol{\phi}$ may be noise,

$$P(\boldsymbol{\phi} \in C^\alpha) = P(\boldsymbol{\phi} \in C^\alpha \mid \exists \beta : \boldsymbol{\phi} \in C^\beta) P(\exists \beta : \boldsymbol{\phi} \in C^\beta) . \tag{9}$$

Here, $P(\exists \beta : \boldsymbol{\phi} \in C^\beta)$ is the probability that $\boldsymbol{\phi}$ belongs to some cluster. To estimate $P(\exists \beta : \boldsymbol{\phi} \in C^\beta)$, the $\lambda$ value at which $\boldsymbol{\phi}$ would join the nearest cluster is compared to the maximum $\lambda$ value of that cluster,

$$P(\exists \beta : \boldsymbol{\phi} \in C^\beta) = \frac{\lambda_{\boldsymbol{\phi} \to C^\alpha}}{\lambda_{\text{max}}^{C^\alpha}} , \tag{10}$$

where $\lambda_{\boldsymbol{\phi}\rightarrow C^\alpha}$ is the persistence value at which point $\boldsymbol{\phi}$ would join its nearest cluster $C^\alpha$ and $\lambda_{\max}^{C^\alpha}$ is the maximum $\lambda$ value of cluster $C^\alpha$. Thus, the final probability, that point $\boldsymbol{\phi}$ belongs to cluster $C^\alpha$ is,

$$P^\alpha(\boldsymbol{\phi}) = \frac{\lambda_{\boldsymbol{\phi}\rightarrow C^\alpha}}{\lambda_{\max}^{C^\alpha}} \cdot M_{\text{norm}}^\alpha(\boldsymbol{\phi})\,. \tag{11}$$

## B   Details on experimental setup

Here, we provide further details on the experiments.

**ViT sources**   We list the URL of each Vision Transformer provided by the timm library [45]:

- FS: `https://huggingface.co/timm/vit_base_patch16_224.augreg_in1k`
- CLIP: `https://huggingface.co/timm/vit_base_patch16_clip_224.openai`
- DINO: `https://huggingface.co/timm/vit_base_patch16_224.dino`
- MAE: `https://huggingface.co/timm/vit_base_patch16_224.mae`

**Hyperparameters for UMAP and HDBSCAN**   We tune hyperparameters of UMAP and HDB-SCAN such that the density-based validity index DBCV is maximized across models and layers. Here, for DBCV, the average across clusters is weighted by their respective size such that the noise rate is indirectly included. We list the effect of the most influential hyperparameters that we tune and state the final value we used:

- **Minimal distance in UMAP**: a low minimal distance in UMAP enhances local cluster density but may also increase noise. We use a value of 0.01 in all experiments.
- **Number of neighbours in UMAP**: the number of neighbors controls the local structure, the smaller the finer it captures local neighborhoods but distorts global structure which is important for concept alignment analysis later. We use a value of 30 in all experiments.
- **Embedding dimensionality in UMAP**: We use the practical limit for HDBSCAN of $F' = 50$ in all experiments.
- **Minimum cluster size in HDBSCAN**: a too small minimum cluster size may identify noise as a cluster, whereas, when too large, distinct clusters will merge. We use a value of 50 in all experiments.
- **Min samples in HDBSCAN**: controls how conservative the algorithm is about noise. We need this to be rather low because of sampling limitations which means that most likely some concept manifolds are not sampled densely enough. We use a value of 20 in all experiments.

Additionally, we assume that clusters are rather uniform in size and select the leaf nodes in the HDBSCAN hierarchical condensed tree as clusters. Sampling one pooled SEQ token or one CLS token from each representation of images within a 25% subset of the ImageNet1-1k train set results in 315.770 feature vectors $\phi_i$ for clustering. We use the cuML [36] versions of HDBSCAN and UMAP for computation on the GPU.

In Tab. 2 and Tab. 3, we show a hyperparameter sensitivity study (at layer 11 of DINO SEQ representations) that varies UMAP and HDBSCAN parameters one at a time (the other hyperparameters are fixed at the default values stated above) and observe that concept discovery evaluation metrics are reasonably stable within a quite broad hyperparameter range. Only NRMSE shows a rather high range when varying the embedding dimension and UMAP minimal distance with minimum of $< 1.0$ for small embedding dimensions ($\leq 40$) and very small minimal distance. This is likely not a sign of a superior embedding, but rather an counter intuitive behavior of this metric due to extreme compression of pairwise distances in the embedding.

**Cluster label in Concept Atlas**   To assign a label from the WordNet Hierarchy to each concept cluster, we first assign the ImageNet-1k label of the image from which a token is extracted to its representation feature vector $\phi_i$. Then we map this to a label higher in the WordNet hierarchy by the mapping in Tab. 4. We then assign the most frequent label among the cluster members $\{\phi_j^\alpha\}$ to the cluster $C^\alpha$.

| Hyperparameter | Range | NRMSE | Noise ratio | DBCV | Robustness |
|---|---|---|---|---|---|
| Embedding dim. $F'$ | $20 - 100$ | $0.7 < 9.5 < 15.1$ | $0.66 < 0.66 < 0.66$ | $0.56 < 0.58 < 0.59$ | $0.84 < 0.84 < 0.87$ |
| UMAP min. dist. | $0.005 - 0.5$ | $0.7 < 9.5 < 13.6$ | $0.64 < 0.66 < 0.72$ | $0.53 < 0.58 < 0.58$ | $0.81 < 0.85 < 0.86$ |
| UMAP neighbors | $10 - 40$ | $6.4 < 9.5$ | $0.65 < 0.66 < 0.66$ | $0.58 < 0.58 < 0.61$ | $0.84 < 0.85 < 0.85$ |

Table 2: Impact of UMAP hyperparameters on embedding and clustering quality at layer 11 of DINO SEQ representations. UMAP parameters are varied one at a time while all other hyperparameters are fixed at the default values stated above. Results are reported as min<default<max.

| Hyperparameter | Range | Noise ratio | DBCV | Robustness |
|---|---|---|---|---|
| Min. cluster size | $20 - 100$ | $0.57 < 0.66 < 0.73$ | $0.57 < 0.58 < 0.59$ | $0.83 < 0.84 < 0.85$ |
| Min. samples | $10 - 50$ | $0.66 < 0.66 < 0.67$ | $0.50 < 0.58 < 0.65$ | $0.83 < 0.84 < 0.85$ |

Table 3: Impact of HDBSCAN hyperparameters on clustering quality at layer 11 of DINO SEQ representations. HDBSCAN parameters are varied one at a time while all other hyperparameters are fixed at the default values stated above. Results are reported as min<default<max.

**Computation of alignment**    Our concept-based alignment measure CBA is based on pairs of feature vectors $(\phi_i, \phi_j)$. To reduce run-time, we sub-sample 20% of the 315.770 feature vectors before computing CBA. Some concepts become under-sampled in this process, indicated by their total activation in the subsample being much lower than 20% of their activation over all samples. Such under-sampled concepts appear close to nearly all other concepts due to the missing activations. To address this, we exclude them by filtering out concepts whose total activation in the subsample is below 17% of their total activation across the entire dataset (this filters out around 4% of concepts in the examples shown here and in the main paper).

**Runtime**    Collecting activations for a 25% subset of the ImageNet1-1k train set, concept discovery and concept discovery evaluation (NRMSE, DBCV) takes around 1.5 hours per layer. We performed around 30 runs for UMAP and HDBSCAN hyperparameter tuning. Computing scalar concept alignment for all pairs of layers depends on the total number of concepts, requiring about 1.25 hours for the FS model, which has the largest concept count. Concept-wise distance computations also scale with the number of concepts: for example, the fine-grained inter-model analysis shown in Fig. 6 in the main paper took about 6 hours on MAE vs. DINO at layer 10 with 477 and 684 concepts, respectively. All experiments were conducted on a Tesla V100 GPU.

## C    Baselines for concept discovery evaluation

Here, we describe two baselines for our evaluation of NLMCD concept discovery quality in Sec. 4.1 of the main paper. We present them alongside the concept discovery results for FS, CLIP, DINO, and MAE in Fig.7.

**Random/ImageNet Baseline**    We feed ImageNet samples (the same subset as in Sec. 4.1) to a randomly initialized ViT. Although the input images themselves contain discriminative patterns, the model itself has no learned structure. Here, noise rate remains almost constant across layers, comparable to middle or later layers of trained models. UMAP preserves distances well in this setting, resulting in good NRMSE. DBCV scores (measuring cluster seperation by density) are overall lower than in trained models but not as low as one might expect. Visually, clusters seem to capture pixel color similarities rather than semantic concepts (see examples in Fig.7). However, robustness of concept discovery is much lower: different initializations of UMAP and HDBSCAN lead to different color-driven clusters. In short, while input structure alone can induce clustering, it mainly reflects superficial color patterns rather than meaningful learned semantics with slightly reduced cluster separation and lower robustness.

**Random/Noise Baseline**    Here, a randomly initialized ViT receives Gaussian noise patches. With no structure in the data or the model weights, HDBSCAN labels most points as noise, leading to a higher noise rate than concept discovery with ImageNet inputs or trained models. Similarly, DBCV values drop noticeably, whereas NRMSE remains only slightly higher, indicating that UMAP still preserves pairwise distances. Interestingly, robustness is relatively high in this scenario, simply because most tokens are consistently categorized as noise. This extreme setting underscores that

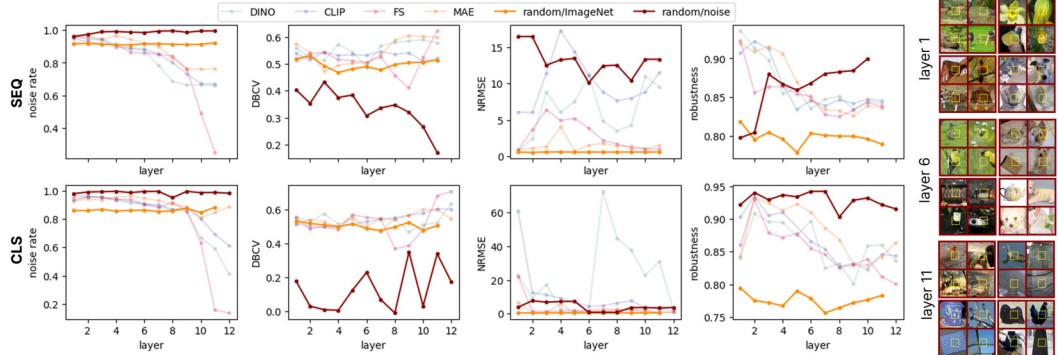

Figure 7: Two baselines for concept discovery evaluation based on a randomly initialized model with ImageNet or gaussian noise input. **Left:** The noise rate is the ratio of points classified as noise in HDBSCAN. DBCV is a density-based clustering validity index that contrasts intra- vs inter-cluster density with scores in $[-1, 1]$ where higher is better. NRMSE measures the root mean squared error between the distance matrix of the original and embedded activations, normalized by the average distance in the original embedding, and shows how faithfully the UMAP embedding captures the geometry of the representation. Robustness is measured between two runs by concept alignment from Eq. 2. Results are across layers for SEQ (upper row) and CLS (bottom row) token representations. **Right:** Randomly chosen examples for concepts discovered at layer 1, 6, 11 in the SEQ representations of the randomly initialized model, all corresponding to color patterns.

both structured data and learned parameters are essential for creating meaningful clusters in concept discovery.

**Summary**   Overall, the Random/ImageNet baseline shows that the input distribution alone can drive clustering based on superficial pixel color regularities in representations, but naturally fails to produce semantically rich clusters driven by the model's learned concepts. The Random/Noise baseline confirms that without both learned parameters and structured data, clustering breaks down almost entirely, highlighting the fundamental role of learned representations for robust and meaningful concept discovery.

## D   Concept Formation Graphs

**Notation.** Let $C^{l,\alpha}$, with $\alpha \in 1, \ldots, N^l$, denote a concept cluster in layer $l$. For a token $i \in 1, \ldots, k$, we use the notation $\boldsymbol{\phi}_i^l \in C^{l,\alpha}$ to indicate that the token representation $\boldsymbol{\phi}_i^l$ in layer $l$ is assigned to that cluster. Let $C^{l,*}$ denote the target concept for which a concept formation graph (CFG) is constructed.

Given $k$ tokens sampled from the training dataset and their cluster assignments, the algorithm for constructing the CFG is defined as follows:

1. **Transition matrix calculation:** First, we compute transition matrices $T_{l,l+1} \in \mathbb{Z}^{N^l \times N^{l+1}}$ for each pair of consecutive layers $(l, l+1)$. Each entry represents the count of tokens transitioning from a concept $C^{l,\alpha}$ in layer $l$ to a concept $C^{l+1,\beta}$ in layer $l+1$:

$$(T_{l,l+1})_{\alpha\beta} = \#\{i \in \{1, \ldots, k\} :$$
$$\boldsymbol{\phi}_i^l \in C^{l,\alpha} \text{ and } \boldsymbol{\phi}_i^{l+1} \in C^{l+1,\beta}\} \qquad (12)$$

   where $\#\{\cdot\}$ denotes the count of tokens.

2. **Recursive graph construction:** Initializing the set of CFG nodes with the target concept node $C^{l,*}$, we recursively add all predecessor concept nodes whose "contribution" (the proportion of incoming transitions) surpasses a specified threshold $\tau$. Formally, suppose a concept $C^{l+1,\beta}$ in layer $l+1$ has been added to the CFG. Then, for each concept $C^{l,\alpha}$ in

layer $l$, we include the edge $(C^{l,\alpha}, C^{l+1,\beta})$ and node $C^{l,\alpha}$ to the CFG if:

$$\frac{(T_{l,l+1})_{\alpha\beta}}{\sum_{\gamma=1}^{N^l} (T_{l,l+1})_{\gamma\beta}} > \tau. \tag{13}$$

The resulting CFG is a binary, unidirectional graph. Fig. 9, 10, and 8 illustrate additional exemplary CFGs for CLIP and DINO. The CFGs were constructed using the same $k = 315{,}770$ tokens from the ImageNet training dataset that were used for concept discovery, with the threshold parameter set to $\tau = 0.05$. In each image, the "concept" token is highlighted in yellow. The concepts in Fig. 3 of the main text are human-labeled.

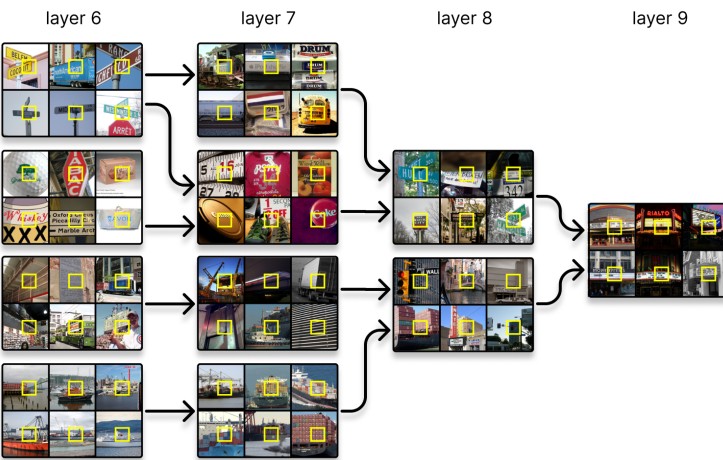

Figure 8: Concept formation graph for a concept in layer 9 of DINO. Each concept is represented by six randomly sampled images containing a token assigned to that concept (highlighted in a yellow frame).

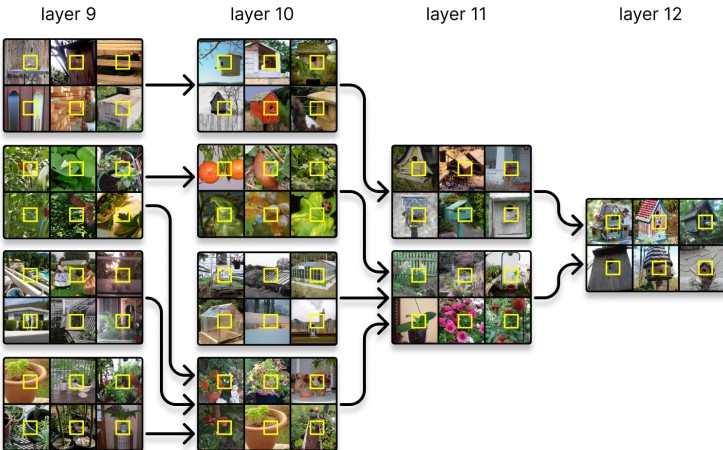

Figure 9: Concept formation graph for a concept in layer 12 of CLIP. Each concept is represented by six randomly sampled images containing a token assigned to that concept (highlighted in a yellow frame).

## E    Qualitative comparison against linear and spherical concept discovery

Figure 11 illustrates the concept atlases for four concept discovery methods (PCA, MCD, KMeans, and NLMCD) in the concept alignment sanity check of Sec. 4.2 in the main paper. For a direct qualitative comparison, we also show concept atlases at layer six for CLIP SEQ token concepts using

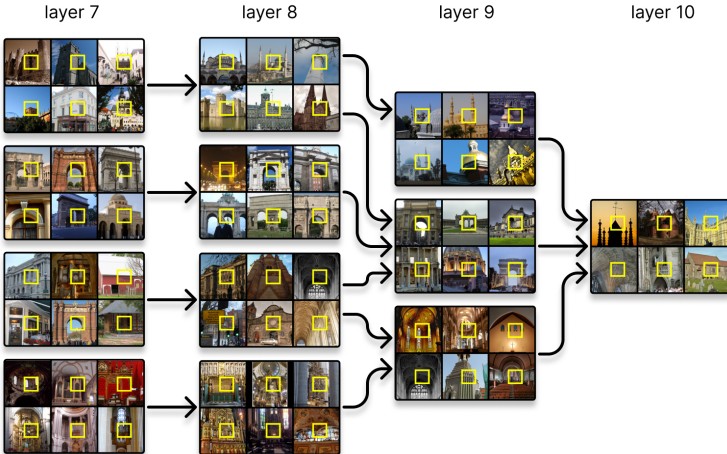

Figure 10: Concept formation graph for a concept in layer 10 of CLIP. Each concept is represented by six randomly sampled images containing a token assigned to that concept (highlighted in a yellow frame).

each method (Fig. 12–15). We segment each atlas into 15 groups via KMeans (indicated by color) and display random examples of concepts within each group.

Visually, MCD, KMeans, and NLMCD exhibit more coherent concept clusters than PCA. Moreover, when examining groups of close concepts in each atlas, we find that NLMCD and KMeans yield more structured organization and greater semantic consistency within concept groups than MCD or PCA (e.g., note how the bright yellow cluster in the MCD atlas mixes sundown/horizon, fur, and shiny object concepts). While the difference in semantic consistency is subtle between KMeans and NLMCD in this example, the results in Sec. 4.2 show that KMeans is on par with NLMCD in half of the test settings but never outperforms it. Nevertheless, KMeans can still serve as a slightly less accurate but more computationally cheap alternative to NLMCD.

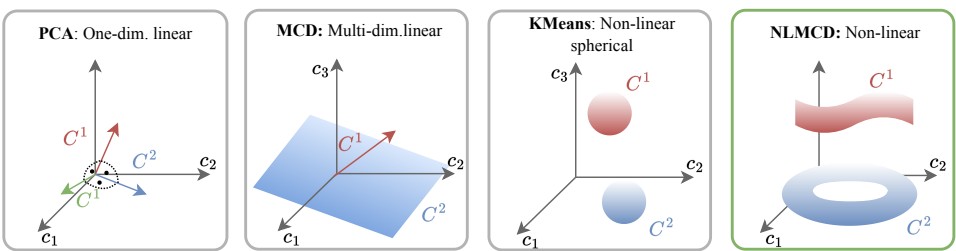

Figure 11: Illustration of the four different concept definitions that are compared for measuring concept alignment in Sec. 4.2 in the main paper (in three dimensions).

## F  Concept alignment analysis

### F.1  CLS representations

We investigate concept-based alignment within and across models based on the CLS token representations analogous to the SEQ token analysis in the main paper.

**Intra-model alignment**   We first compare the intra-model alignment heatmaps for CLS representations across layers (top row of Fig. 16), measured by CBA, with the same analysis for SEQ tokens presented in the main paper. For the FS model, the CLS representations exhibit a pattern very similar to that of the SEQ representations. Likewise, CLIP and MAE show largely comparable CLS–SEQ alignment profiles, but their earliest layers display noticeably lower alignment for CLS,

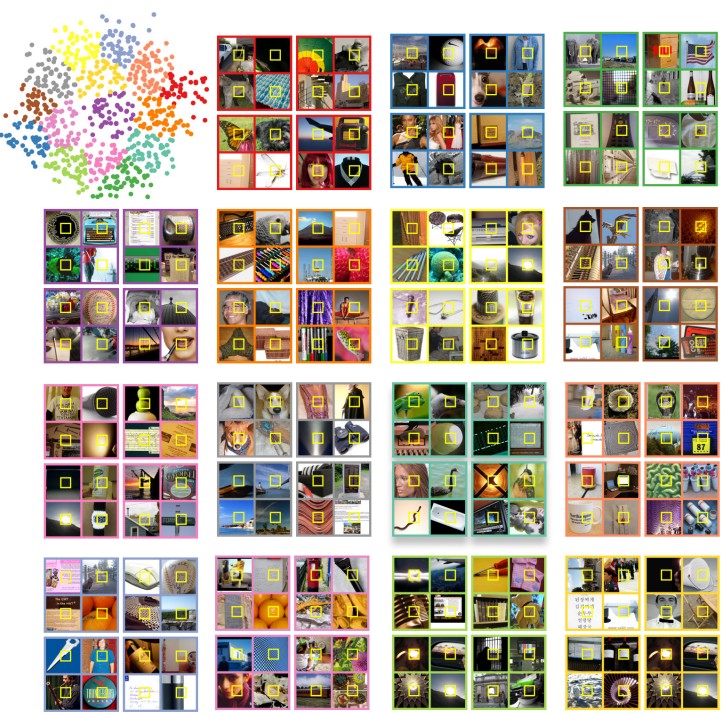

Figure 12: Detailed concept atlas for PCA SEQ token concepts at layer six of CLIP. This is based on a UMAP embedding constructed from the pairwise distance of concepts measured by $d_{cross}(P^\alpha, P^\beta)$. Each point in this concept atlas corresponds to a distinct concept $P^\alpha$. To convey their meaning, we show four random input tokens from the members of the concept cluster $P^\alpha$. We dissect the concept atlas into 15 groups (indicated by color) and show four random concepts for each group.

suggesting that the model might rely less on the CLS token in early layers. With respect to class label alignment, intrinsic dimensionality, and the number of discovered concepts, the CLS patterns in Fig. 17 also broadly resemble the SEQ results, except for DINO. In DINO's CLS tokens, alignment across layers is substantially lower than in the SEQ tokens, and the concept atlas in layer 11 appears less semantically organized. One plausible explanation is that the UMAP embeddings for DINO's CLS tokens (layers 7–11) exhibit higher NRMSE, indicating reduced faithfulness of the embedding to the original activations, and thereby less faithful concepts.

**Inter-model alignment**    Second, we analyze how the CLS representations between two different models differ and present CBA alignment between all layers and model in the upper part of 18. Like for the SEQ representations, CLS representations at layers of the first are more aligned than those of the second half across all models pairs, suggesting that basic foundational features are learned similarly across models, while later layers diverge as the models specialize to concepts serving their pre-training task. However, the overall alignment between models is weaker for CLS representations than for SEQ, also in low layers. Next, we zoom in into the distance $d_{cross}(P^\alpha, P^\beta)$ between concept pairs from CLS representations of MAE layer 11 and CLIP layer 11 in the right part of 18. Next to the pairwise distance matrix $d_{cross}(P^\alpha, Q^\beta)$, we present the distribution of the distances between nearest matches, and randomly chosen examples of concepts from MAE and their nearest matches in CLIP. Similar to the comparison of SEQ concepts between MAE and DINO in Sec. 4.3 in the main paper, we find that 80% of CLS concepts in MAE are similar to only 5 *singular* concepts in CLIP. Lastly, among the randomly chosen examples, visual discrepancy is more pronounced for high-distance concepts than for the other pairs with lower distance.

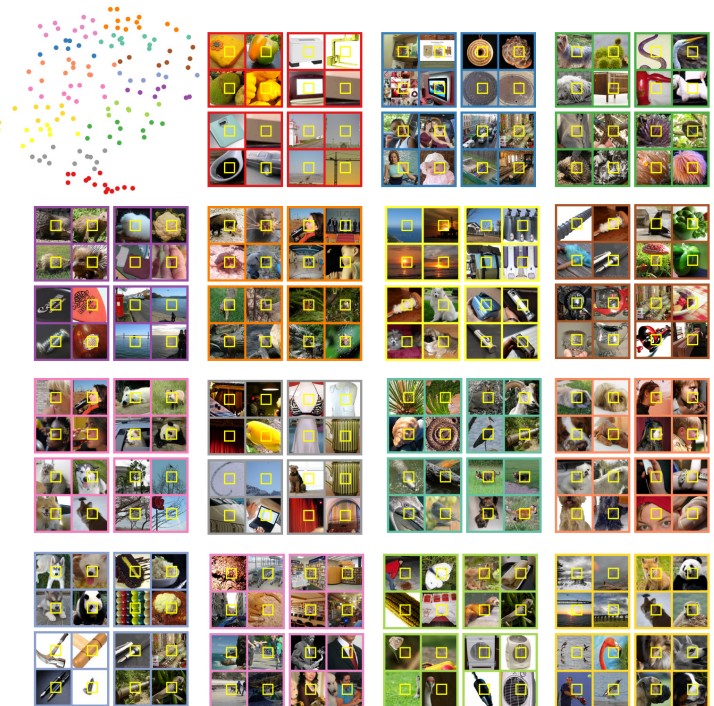

Figure 13: Detailed concept atlas for MCD SEQ token concepts at layer six of CLIP. Detailed concept atlas for PCA SEQ token concepts at layer six of CLIP. This is based on a UMAP embedding constructed from the pairwise distance of concepts measured by $d_{cross}(P^\alpha, P^\beta)$. Each point in this concept atlas corresponds to a distinct concept $P^\alpha$. To convey their meaning, we show four random input tokens from the members of the concept cluster $P^\alpha$. We dissect the concept atlas into 15 groups (indicated by color) and show four random concepts for each group.

## F.2 Additional results for SEQ representations

**Intra-model** To give a more detailed view of the organization of concepts across the layers of one model, we select the DINO model and show the respective concept atlases at layer one, six and eleven in 19, 20, and 21, respectively. To give an overview of the structure within a concept atlas, we group the concepts in the UMAP embedding via KMeans and show four random concepts for each group. In layer one, many concepts correspond to color, in layer six, they represent mostly textures, and in layer eleven they correspond to abstract concepts. Moslty, concepts within a group are of similar nature.

**Inter-model** In the main paper, we show fine-grained inter-model concept distances between DINO and MAE at layer ten. Here, we add fine-grained concept distance analysis between DINO and MAE, as well as between MAE and CLIP in 22. Again, we show the full pairwise distance matrix as well as how distances between closest matching pairs are distributed. Similar to the comparison of SEQ concepts between MAE and DINO in Sec. 4.3 in the main paper, we find that in the case of DINO vs. CLIP, 65% of CLIP concepts most closely match with only 2 *singular* concepts in DINO; for MAE vs. CLIP, 86% of MAE concepts match with 4 concepts in CLIP.

Now, we examine differences in concept alignment between pairs of layers. Specifically, we compare layer 10 in MAE/DINO (see Fig. 6) and the same layer in CLIP/DINO (see Fig. 16), which exhibit the same overall concept alignment (0.74). To summarize fine-grained differences in concept-alignment, we group concept nearest-neighbor-pairs into the WordNet categories from Fig. 4 (88% of concept pairs are in the same category for MAE/DINO, 93% for CLIP/DINO). We then compute the mean concept distance within each concept category and find that 1) canine and insect concepts have high distance in both model pairs, while equipment and material concept have low distance in both pairs; 2) fish concepts are similar in MAE/DINO but have high distance in CLIP/DINO, while sport concepts

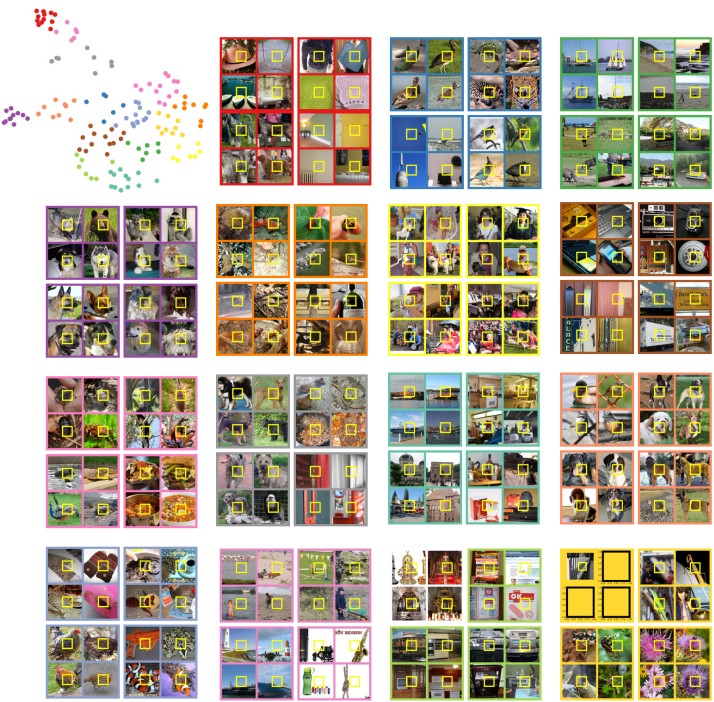

Figure 14: Detailed concept atlas for KMeans SEQ token concepts at layer six of CLIP. Detailed concept atlas for PCA SEQ token concepts at layer six of CLIP. This is based on a UMAP embedding constructed from the pairwise distance of concepts measured by $d_{cross}(P^\alpha, P^\beta)$. Each point in this concept atlas corresponds to a distinct concept $P^\alpha$. To convey their meaning, we show four random input tokens from the members of the concept cluster $P^\alpha$. We dissect the concept atlas into 15 groups (indicated by color) and show four random concepts for each group.

show the opposite trend. This exemplifies a scenario where two representations are similarly aligned by the overall score, but fine-grained concept alignment differs.

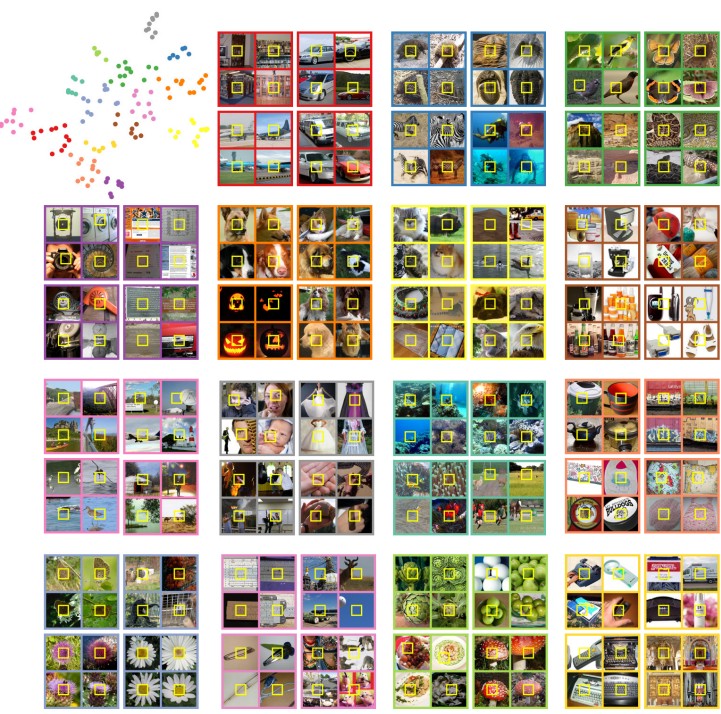

Figure 15: Detailed concept atlas for NLMCD SEQ token concepts at layer six of CLIP. Detailed concept atlas for PCA SEQ token concepts at layer six of CLIP. This is based on a UMAP embedding constructed from the pairwise distance of concepts measured by $d_{cross}(P^\alpha, P^\beta)$. Each point in this concept atlas corresponds to a distinct concept $P^\alpha$. To convey their meaning, we show four random input tokens from the members of the concept cluster $P^\alpha$. We dissect the concept atlas into 15 groups (indicated by color) and show four random concepts for each group.

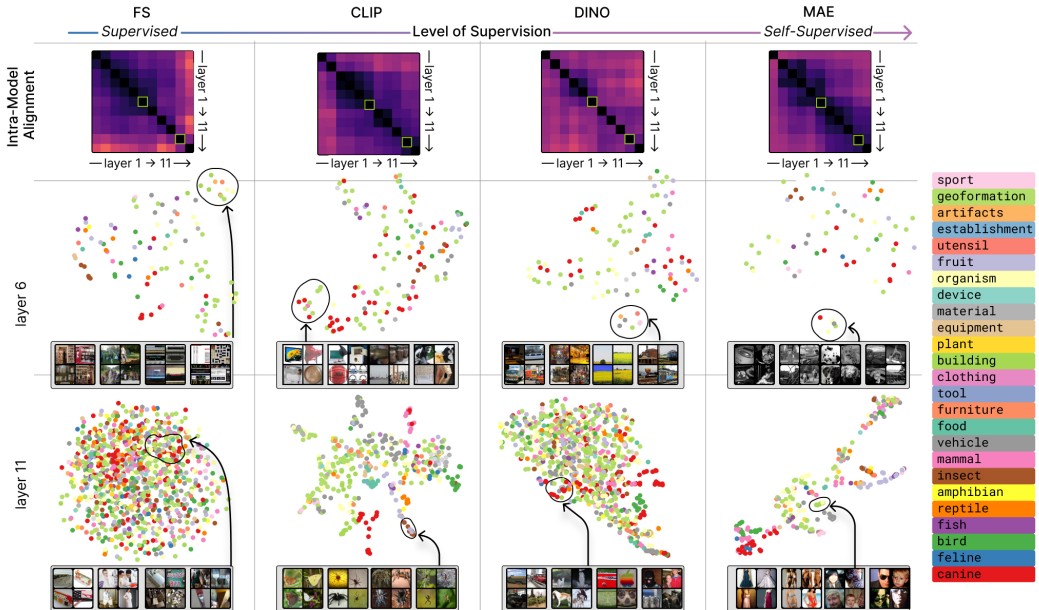

Figure 16: Intra-model relationships based on CLS representations across layers. In the **upper row**, we show CBA to visualize how representations are transformed across layers of the models (darker pixels correspond to higher alignment). In the **center and bottom row** we zoom into the representations at layer 6 and 11 of each model and partition the scalar CBA alignment into single concepts. We show a UMAP embedding constructed from the pairwise distance of concept measured by $d_{cross}(P^{\alpha}, P^{\beta})$. Each point in this *concept atlas* corresponds to a distinct concept $P^{\alpha}$. To convey their meaning, we show four random input tokens from the members of the concept cluster $C^{\alpha}$ marked by a yellow box in the entire image.

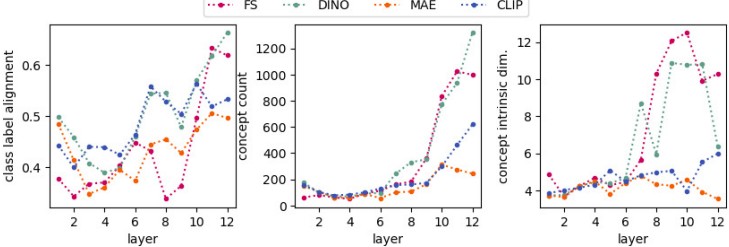

Figure 17: Class label alignment (based on CBA), concept count, and the average intrinsic dimensionality (based on [13]) across concepts for CLS representations supplement the intra-model alignment analysis, by providing insights into how well the model aligns with ImageNet-1k labels, the spatial organization of tokens, and the complexity of the learned concepts as they evolve through the layers.

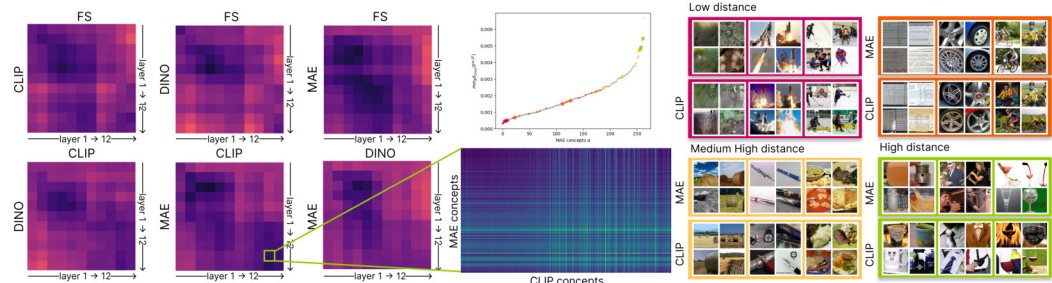

Figure 18: Inter-model relationships based on CLS representations across layers. **Right:** We show CBA to visualize how representations differ between the models (darker pixels correspond to higher alignment). **Left:** We zoom in into the concept-wise distances $d_{cross}(P^\alpha, P^\beta)$ between the representation of in MAE and CLIP at layer 11. Alongside the pairwise distance matrix $d_{cross}(P^\alpha, Q^\beta)$, we present randomly chosen examples of concepts from MAE and their nearest matches in CLIP, and the distribution of the distances between nearest matches. We illustrate some random example pairs with low to high distance (marked by a star in the distance distribution plot).

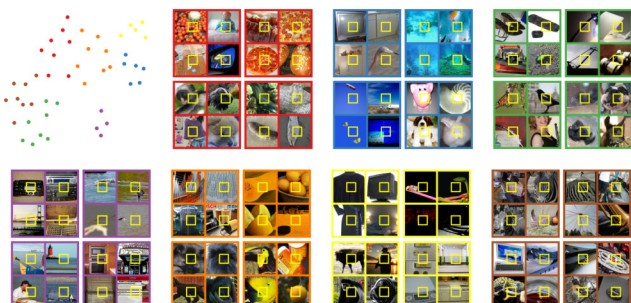

Figure 19: We zoom into the SEQ representations at DINO layer 1 and show a UMAP embedding constructed from the pairwise distance of concepts measured by $d_{cross}(P^\alpha, P^\beta)$. Each point in this *concept atlas* corresponds to a distinct concept $P^\alpha$. To convey their meaning, we show four random input tokens from the members of the concept cluster $P^\alpha$. We dissect the concept atlas into 7 groups and show four random concepts for each group. Concepts representing similar colors lie within the same group, e.g. shades of blue in the blue group or red and orange in the red group.

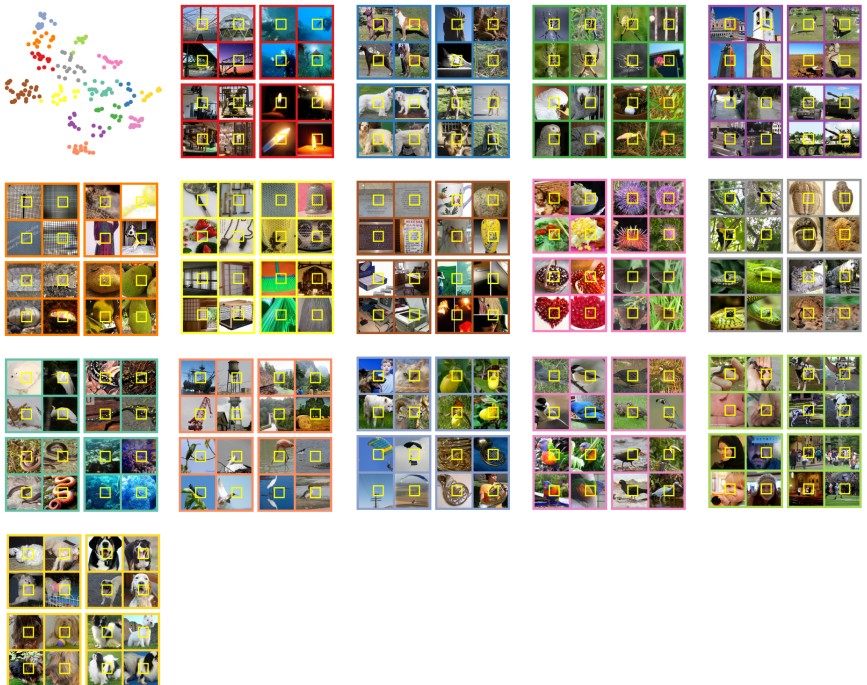

Figure 20: We zoom into the SEQ representations at DINO layer 6 and show a UMAP embedding constructed from the pairwise distance of concepts measured by $d_{cross}(P^\alpha, P^\beta)$. Each point in this *concept atlas* corresponds to a distinct concept $P^\alpha$. To convey their meaning, we show four random input tokens from the members of the concept cluster $P^\alpha$. We dissect the concept atlas into 15 groups and show four random concepts for each group. Most concepts represent a pattern or texture which are similar within each group.

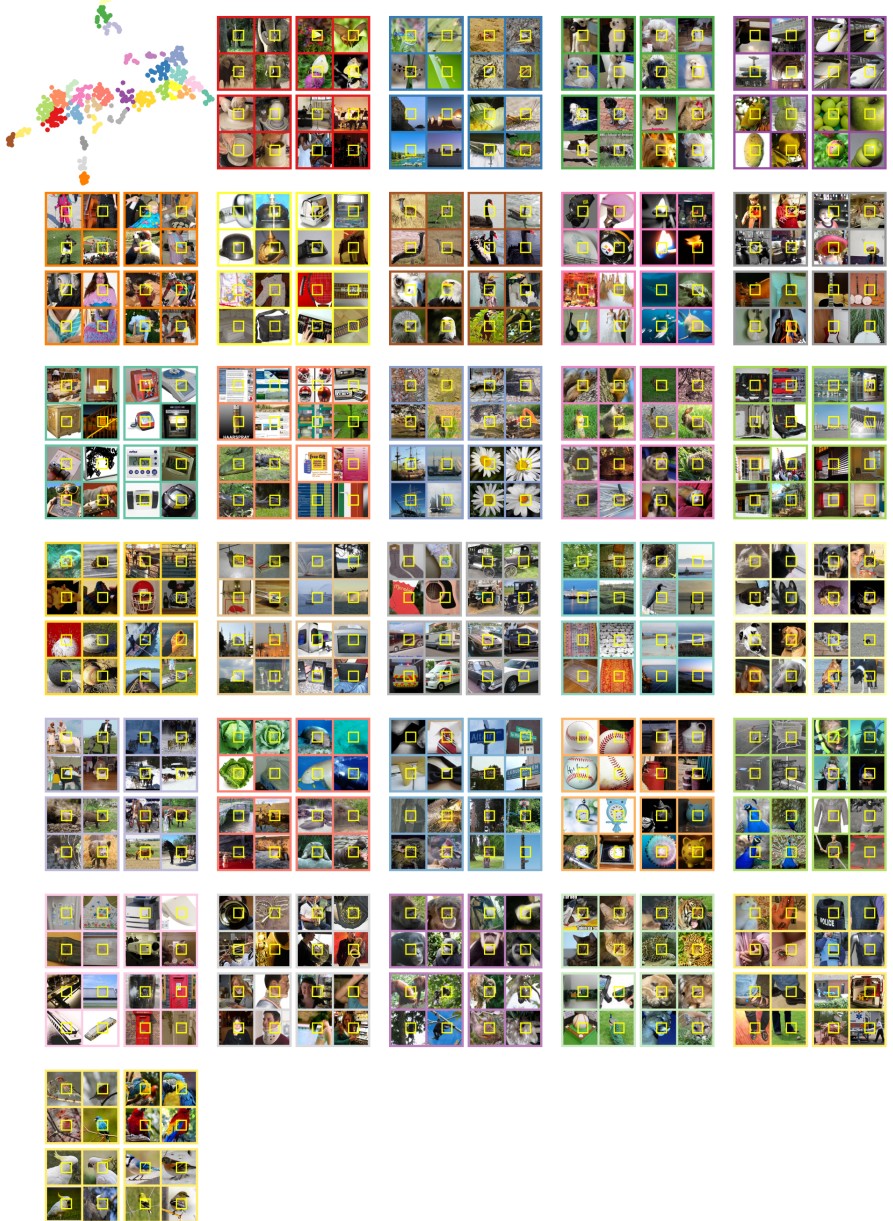

Figure 21: We zoom into the SEQ representations at DINO layer 11 and show a UMAP embedding constructed from the pairwise distance of concepts measured by $d_{cross}(P^\alpha, P^\beta)$. Each point in this *concept atlas* corresponds to a distinct concept $P^\alpha$. To convey their meaning, we show four random input tokens from the members of the concept cluster $P^\alpha$. We dissect the concept atlas into 30 groups and show four random concepts for each group. For most groups, these are semantically similar.

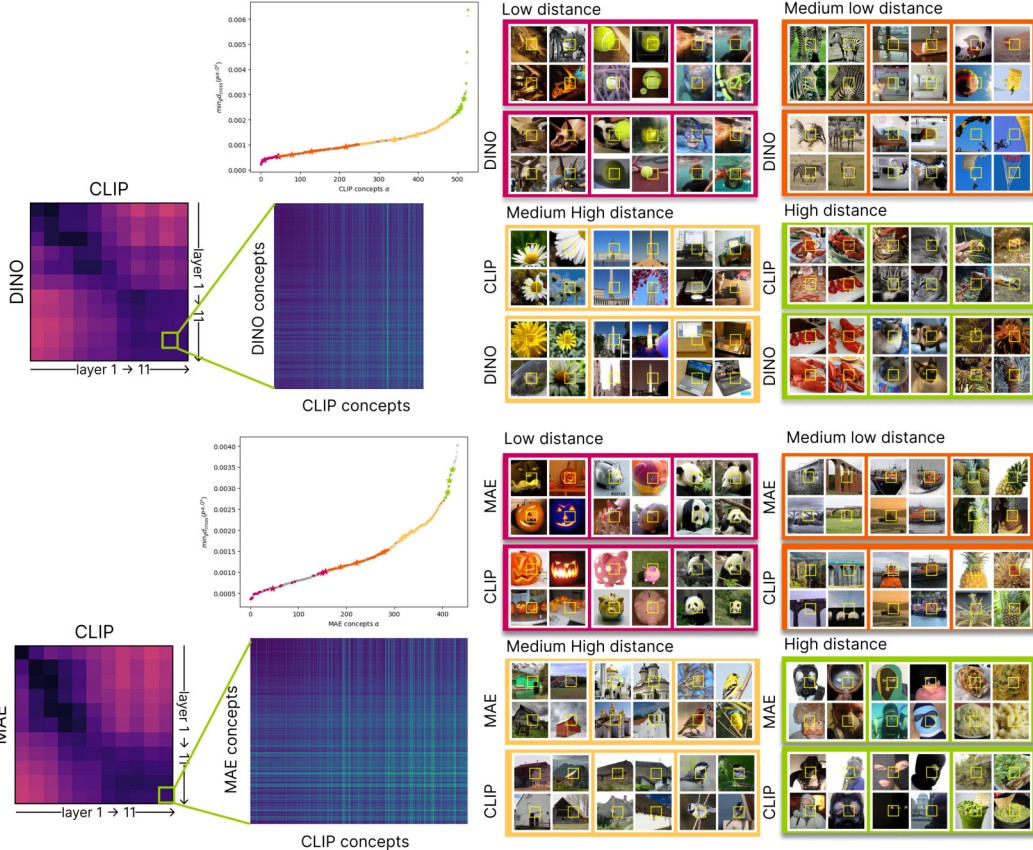

Figure 22: CBA of SEQ concepts across layers of CLIP and DINO, as well as CLIP and MAE (darker pixels correspond to higher alignment). We zoom in into the concept-wise distances $d_{cross}(P^\alpha, P^\beta)$ between the representation of layer ten in CLIP and DINO, as well as layer 11 in CLIP and MAE. Alongside the pairwise distance matrix $d_{cross}(P^\alpha, Q^\beta)$, we present randomly chosen examples of concepts from MAE and their nearest matches in DINO/CLIP, and the distribution of the distances between nearest matches. We illustrate some random example pairs with low to high distance (marked by a star in the distance distribution plots).

| Category | ImageNet-1k class |
| --- | --- |
| amphibian | European fire salamander axolotl bullfrog common newt eft spotted salamander tailed frog tree frog |
| artifacts | Afghan hound Band Aid Dutch oven Petri dish abacus ashcan backpack ballpoint bannister barrel bath towel bathtub beacon beaker beer bottle beer glass bell cote binder birdhouse book jacket bottlecap brass breakwater breastplate broom bucket cannon carousel carton cassette chain mail chainlink fence chiffonier cleaver cliff dwelling cloak clog cocktail shaker coffee mug comic book cowboy boot crate crib crutch cuirass cup diaper dishwasher dock envelope espresso maker face mug fig fire screen flagpole fountain fountain pen gasmask goblet grasshopper grille grocery store guillotine hair spray hand blower holster honeycomb iron "jack-o'-lantern" joystick ladle lampshade lens cap library lipstick lotion mailbag mailbox manhole cover mask matchstick maze measuring cup megalith menu microwave minibus mixing bowl mobile home mortar mortarboard mosquito net mountain tent muzzle necklace obelisk packet paddle patio pedestal pencil box pencil sharpener perfume pickelhaube picket fence pier piggy bank pill bottle pillow pitcher plastic bag plate rack pole pop bottle pot prayer rug purse quill quilt racket radio rain barrel refrigerator rotisserie rubber eraser running shoe safe saltshaker scabbard school bus schooner scoreboard shield shoji shopping basket shower curtain ski mask sleeping bag sliding door soap dispenser soup bowl space bar spotlight steel arch bridge stone wall stove street sign stretcher sunscreen suspension bridge swab swing teddy television thatch theater curtain thimble tile roof totem pole traffic light tray triumphal arch trolleybus tub turnstile umbrella vacuum vase viaduct waffle iron washbasin washer water bottle water jug water tower web site whiskey jug window screen window shade wine bottle worm fence wreck yurt |
| bird | African grey American coot American egret European gallinule albatross bald eagle bee eater bittern black grouse black stork black swan brambling bulbul bustard chickadee cock coucal crane dowitcher drake flamingo goldfinch goose great grey owl hen hornbill house finch hummingbird indigo bunting jacamar jay junco king penguin kite limpkin little blue heron lorikeet macaw magpie ostrich oystercatcher partridge pelican prairie chicken ptarmigan quail red-backed sandpiper red-breasted merganser redshank robin ruddy turnstone ruffed grouse spoonbill sulphur-crested cockatoo toucan vulture water ouzel white stork |
| building | apiary barn boathouse castle church cinema greenhouse home theater monastery mosque palace planetarium prison restaurant stage stupa vault |
| canine | African hunting dog Airedale American Staffordshire terrier Appenzeller Arctic fox Australian terrier Bedlington terrier Bernese mountain dog Blenheim spaniel Border collie Border terrier Boston bull Bouvier des Flandres Brabancon griffon Brittany spaniel Cardigan Chesapeake Bay retriever Chihuahua Dandie Dinmont Doberman English foxhound English setter English springer EntleBucher Eskimo dog French bulldog German short-haired pointer Gordon setter Great Dane Great Pyrenees Greater Swiss Mountain dog Ibizan hound Irish setter Irish terrier Irish water spaniel Irish wolfhound Italian greyhound Japanese spaniel Kerry blue terrier Labrador retriever Lakeland terrier Leonberg Lhasa Maltese dog Mexican hairless Newfoundland Norfolk terrier Norwegian elkhound Norwich terrier Pekinese Pembroke Pomeranian Rhodesian ridgeback Rottweiler Saint Bernard Saluki Samoyed Scotch terrier Scottish deerhound Sealyham terrier Shetland sheepdog Shih-Tzu Siberian husky Staffordshire bullterrier Sussex spaniel Tibetan mastiff Tibetan terrier Walker hound Weimaraner Welsh springer spaniel West Highland white terrier Yorkshire terrier affenpinscher basenji basset beagle black-and-tan coonhound bloodhound bluetick borzoi briard bull mastiff cairn chow clumber cocker spaniel collie coyote curly-coated retriever dalmatian dhole dingo flat-coated retriever giant schnauzer golden retriever grey fox groenendael hyena keeshond kelpie kit fox komondor kuvasz malamute malinois miniature pinscher miniature poodle miniature schnauzer otterhound papillon pug red fox red wolf redbone schipperke silky terrier soft-coated wheaten terrier standard poodle standard schnauzer timber wolf toy poodle toy terrier vizsla whippet white wolf wire-haired fox terrier |
| clothing | Christmas stocking Loafer Old English sheepdog Windsor tie abaya academic gown apron bathing cap bearskin bib bikini bolo tie bonnet bow tie brassiere bulletproof vest cardigan chest cowboy hat crash helmet dishrag feather boa fur coat gown handkerchief hook hoopskirt jean jersey kimono knee pad lab coat maillot military uniform miniskirt mitten overskirt pajama paper towel poncho sandal sarong seat belt shower cap sock sombrero stole suit sweatshirt swimming trunks trench coat velvet vestment wallet wig wool |
| device | accordion acoustic guitar analog clock assault rifle banjo barometer bassoon binoculars bow buckle bullet train candle car mirror car wheel cash machine cello chime combination lock desktop computer digital clock digital watch disk brake drum drumstick electric fan electric guitar flute gas pump gong hair slide hammer hamper hand-held computer hard disc harmonica harp hatchet horn hourglass laptop lighter loudspeaker loupe magnetic compass maraca marimba maypole microphone missile monitor mouse mousetrap neck brace notebook oboe odometer oil filter organ oxygen mask paddlewheel padlock paintbrush panpipe parking meter pick "potters wheel" projector puck radiator radio telescope remote control revolver rifle safety pin sax scale screen sewing machine ski slide rule slot slug snorkel solar dish space heater spider web steel drum stethoscope stopwatch strainer sundial sunglass sunglasses switch syringe thresher toaster torch tripod trombone typewriter keyboard upright vending machine violin wall clock whistle |
| equipment | CD player Polaroid camera balance beam barbell "carpenters kit" cassette player cellular telephone computer keyboard croquet ball crossword puzzle dial telephone drilling platform dumbbell golf ball golfcart horizontal bar iPod jigsaw puzzle modem oscilloscope parachute parallel bars pay-phone photocopier ping-pong ball plate punching bag reel reflex camera soccer ball tape player |
| establishment | bakery barbershop bookshop butcher shop confectionery shoe shop tobacco shop toyshop |
| feline | Egyptian cat Persian cat Siamese cat catamount cheetah coil cougar leopard lion panther snow leopard tabby tiger tiger cat |
| fish | anemone fish barracouta coho eel electric ray gar goldfish great white shark hammerhead lionfish puffer rock beauty stingray sturgeon tench tiger shark |
| food | French loaf bagel burrito carbonara cheeseburger chocolate sauce consomme cucumber dough eggnog espresso guacamole hay hot pot hotdog ice cream ice lolly mashed potato meat loaf pizza potpie pretzel red wine trifle |
| fruit | Granny Smith acorn buckeye hip jackfruit lemon orange pineapple rapeseed strawberry |
| furniture | altar barber chair bassinet beaver bookcase china cabinet cradle desk dining table entertainment center file folding chair four-poster medicine chest milk can mink otter park bench pool table rocking chair studio couch table lamp throne toilet seat wardrobe |
| geological formation | alp bubble cliff coral reef dome geyser lakeside promontory sandbar seashore valley volcano |
| insect | ant bee cabbage butterfly cicada cricket damselfly dragonfly dung beetle fly ground beetle lacewing ladybug leaf beetle leafhopper long-horned beetle lycaenid mantis monarch peacock rhinoceros beetle ringlet sulphur butterfly tiger beetle walking stick weevil |
| mammal | African elephant American black bear Angora Arabian camel Indian elephant Madagascar cat Sus scrofa armadillo baboon bighorn bison black-footed ferret brown bear capuchin chimpanzee colobus dugong echidna fitch fox squirrel gazelle gibbon gorilla grey whale guenon guinea pig hamster hare hartebeest hippopotamus ibex ice bear impala indri killer whale koala langur lesser panda llama macaque marmoset marmot meerkat mongoose orangutan ox panda patas platypus polecat porcupine proboscis monkey ram sea lion siamang sloth bear spider monkey squirrel monkey three-toed sloth titi tusker wallaby warthog water buffalo wombat wood rabbit zebra |
| material | chain cornet doormat groom knot spindle toilet tissue |
| musical | grand piano |
| organism | American lobster Dungeness crab German shepherd admiral agaric badger ballplayer barn spider black and gold garden spider black widow bolete boxer brain coral centipede chiton cockroach conch coral fungus crawfish dam ear earthstar fiddler crab flatworm garden spider gyromitra harvester harvestman hen-of-the-woods hermit crab hog howler monkey isopod jellyfish king crab mushroom nematode nipple printer rock crab rule scorpion scuba diver sea cucumber sea slug sea urchin snail spiny lobster starfish stinkhorn tarantula tick trilobite weasel wing wolf spider |
| plant | acorn squash artichoke banana bell pepper broccoli butternut squash cardoon cauliflower corn custard apple daisy head cabbage ocarina pinwheel pomegranate sea anemone sorrel spaghetti squash "yellow ladys slipper" zucchini |
| reptile | African chameleon African crocodile American alligator American chameleon Gila monster Indian cobra Komodo dragon agama alligator lizard banded gecko boa constrictor box turtle common iguana diamondback frilled lizard grass snake green lizard green mamba green snake hognose snake king snake leatherback turtle loggerhead mud turtle night snake ringneck snake rock python sand viper sea snake sidewinder terrapin thunder snake triceratops vine snake water snake whiptail |
| sport | baseball basketball football helmet rugby ball tennis ball volleyball |
| tool | can opener chain saw corkscrew lawn mower letter opener lumbermill nail plane plow plunger power drill screw screwdriver shovel |
| utensil | Crock Pot caldron coffeepot frying pan spatula teapot wok wooden spoon |
| vehicle | Model T aircraft carrier airliner airship ambulance amphibian balloon barrow beach wagon bicycle-built-for-two bobsled cab canoe catamaran chambered nautilus container ship convertible dogsled electric locomotive fire engine fireboat forklift freight car garbage truck go-kart gondola half track horse cart jeep jinrikisha lifeboat limousine liner minivan moped motor scooter mountain bike moving van oxcart passenger car pickup pirate racer recreational vehicle shopping cart snowmobile snowplow space shuttle speedboat sports car steam locomotive streetcar submarine tank tow truck tractor trailer truck tricycle trimaran unicycle wagon warplane yawl |

Table 4: Mapping between categories from the WordNet Hierarchy and the ImageNet-1k classes used for assigning a category to the concept clusters.

