# OpenReview forum: "Beyond Scalars: Concept-Based Alignment Analysis in Vision Transformers"
_NeurIPS.cc/2025/Conference — NeurIPS 2025 poster_

### Official Review · Reviewer_wzkQ · 2025-06-28

**Clarity:** 3
**Significance:** 2
**Originality:** 2
**Rating:** 4
**Confidence:** 5

**Summary:**

This paper proposes a new method for evaluating the similarity and differences in vision models' representation by combining concept discovery with alignment analysis. As opposed to defining concepts as linear subspaces in representation space, this paper approaches concepts as being embedded in low-dimensional non-linear manifolds. Network's internal features are projected via UMAP and then clustered with HDBSCAN. Then, the distances of clusterings from different networks are calculated. This method is evaluated on multiple vision models (e.g., DINO, CLIP, etc.) with ImageNet images.

**Questions:**

* The paper can be further strengthened by showing how the method enables new insights into the representations of these models. Most of the experiments just demonstrate sanity checks. The finding that models supervised with labels have reduced semantic structure in the learned representations is interesting. Why does this not happen in CLIP as it has supervision in the form of text?

Here are a few questions that this method can be used to answer:
* What is the same and different between generative and discriminative models. How does the semantic structure differ and how is it similar?
* I do not see any reason this method cannot be applied to a language model. How is the semantic structure different between the vision and language encoder in CLIP?

These are just a few ideas that can make me more convinced of the utility of the method.

**Ethical Concerns:**

["NO or VERY MINOR ethics concerns only"]

**Final Justification:**

I was fairly borderline on this paper for two main reasons. The first one is novelty, since the paper combines existing tools such as UMAP and HDBSCAN. This method, as acknowledged by the paper, is not very scalable, which limits its applicability on larger models trained on more diverse data, where explainability methods would be more impactful and yield more meaningful insights. While the proposed definition — "a novel definition of concepts as nonlinear manifolds" (L45) — is a unique direction, definitional novelty alone is difficult to assess unless it leads to new types of insight or analysis. Which leads to my second main concern, that the method has not demonstrated to provide a practical application, theory, or produce new scientific insights beyond what has already been shown in explainability/interpretability/representational alignment literature (e.g., earlier layers focus on "lower-level structures, shapes, and object parts" (L270-271)). The analysis is also ImageNet-centric and focuses on only discriminative models, limiting its scope. In my original review, I provided multiple potential applications and questions that the method could be used for, which were **not** resolved in the rebuttal. I am confident in this assessment as I work on representational alignment and interpretability. That being said, the paper is well-executed and the method is fairly well-motivated. I will give the benefit-of-the doubt and maintain my score.

As a sidenote, I was note able to get to this during the discussion. The finding that ViTs supervised with labels have reduced semantic structure in the learned representations is in line with the well-known phenomenon of "neural collapse" [1], where representations in the penultimate layer of a supervised model collapses to the class mean in order to encourage max-margin separation. I think that some discussion of this would be valuable.

[1] Papyan et al. "Prevalence of neural collapse during the terminal phase of deep learning training." PNAS 2020.

**Limitations:**

Yes, the authors have a dedicated limitations subsection mentioning the limited scalability of HDBSCAN.

**Quality:**

3

**Strengths And Weaknesses:**

**Strengths**
* The idea of measuring similarity/differences between networks using interpretable concepts rather than a single scalar is interesting and can be helpful in better probing what these model do and don't know.
* Most works on interpreting representations assume the linear representation hypothesis, while this work is more unique in assuming non-linearity and leveraging non-linear tools.
* The proposed method (NLMCD) passes sanity checks better than baselines. Additionally, some qualitative observations are made the show the utility of NLMCD.

**Weaknesses**
* The ultimate goal of a new alignment metric would be to facilitate some new insight. Examples from the paper, such as finding that earlier layers focus on lower level features, are good sanity checks that have already been established. Some findings that would not be enabled by other similarity metrics would strengthen the paper. The most interesting finding was that models which were supervised with labels have reduced semantic structure in the learned representations. I think this can be emphasized more and further developed.
* The method itself is not completely novel. It combines existing tools such as UMAP and HDBSCAN. This makes it more important to emphasize how this combination of tools enables the discovery of new phenomena which otherwise would not have been discovered.
* [1] seems similar in spirit. It clusters neurons across vision models based on similarity, forming concepts.

[1] Dravid et al. "Rosetta Neurons: Mining the Common Units in a Model Zoo." ICCV 2023.

---

> ### Author Rebuttal · Authors · 2025-07-31
>
> Thank you for your valuable feedback and positive assessment of our method as helpful and unique.
>
> **[W1] New insights**
>
> We agree that this is our most interesting finding, which we highlight it in the abstract and introduction.
> Regarding new insights, we will add a small analysis that compares two representation pairs with same overall alignment but different alignment on the concept level to the appendix, which we describe in further detail in the answer to [Q2] of ErWZ.
>
> **[W2] Novelty**
>
> We agree that the components (UMAP, HDBSCAN) of our concept discovery pipeline are well established. Our contributions, which we describe in the Introduction, rather lie in: (1) the novel definition of concepts as nonlinear manifolds, and (2) decomposing alignment into concept-level comparisons and thereby enabling insights that single-scalar methods like CKA miss.
>
> **[W3] Related work**
>
> Thanks pointing out the paper by Dravid et al., we will add it to our Related Works.
>
> **[Q1] FS vs CLIP model**
>
> The CLIP model differs from fully supervised ViTs in that its supervision is soft and semantically structured (via textual description of images that vary across samples from one class). This encourages representational similarity between fine-grained categories (e.g., all dogs like "Labrador", "German shepherd"), in contrast to hard class separation in fully supervised setups.
>
> **[Q2] Generative vs. discriminative models**
>
> This is an interesting question - we think that Fig. 4 points to some answers for this question by comparing the FS model with MAE, which is trained with a BERT style masked patch reconstruction objective.
>
>
> **[Q3] CLIP text encoder**
>
>
> We agree that cross modal alignment is a natural extension, and NLMCD CBA is directly applicable to representation of the CLIP text encoder Extending this to text and image encoders that were not jointly trained (like Maniparambil et al. 2024 and Huh et al 2024 )
> is also interesting question, to which NLMCD-CBA could add by understanding alignment on a concept level.
>
> Due to the time constraints of the rebuttal period, we defer these experiments to future work. In such an analysis, we would focus on CLS token representations, as image localized concepts (e.g., a cat’s whiskers in an image) do not directly correspond to textual mentions (e.g., “A cat sitting on a table with whiskers.”). We hypothesize that cross modal concept alignment is most meaningful at the final layers, where image representations are more abstract and more likely to align with the concepts described in the associated text.
>
> Mayug Maniparambil, Raiymbek Akshulakov, Yasser Abdelaziz Dahou Djilali, Mohamed El Amine Seddik, Sanath Narayan, Karttikeya Mangalam, Noel E. O'Connor; Proceedings of the IEEE/CVF Conference on Computer Vision and Pattern Recognition (CVPR), 2024, pp. 14334-14343
>
> Minyoung Huh, Brian Cheung, Tongzhou Wang, Phillip Isola
> Proceedings of the 41st International Conference on Machine Learning, PMLR 235:20617-20642, 2024.

---

> ### Comment · Reviewer_wzkQ · 2025-08-03
>
> Thank you for engaging in the discussion! I discuss some points below.
>
> **"(1) the novel definition of concepts as nonlinear manifolds"**
> I would disagree that this definition is particularly novel. For instance, [1] defines concepts as lying on "tightly circumscribed manifolds."
>
> **(2)...decomposing alignment into concept-level comparisons and thereby enabling insights that single-scalar methods like CKA miss."**
> [2] also uses a dictionary-based approach that can provide insight into how visual concepts are represented differently among various models, thus providing concept-level comparisons.
>
> I think that it is fine that this work builds upon previous methods, but my main concern is that the method has not demonstrated to provide a practical utility, theory, or produce new scientific insights beyond what has already been shown in explainability/interpretability/representational alignment literature. The rebuttal has not adequately addressed this concern. However, I think the paper is well-written and organized. I will give the benefit-of-the doubt and maintain my score.
>
> As a sidenote, MAE is not a generative model in the standard sense—it is a self-supervised model trained with a reconstruction loss, but it does not generate new samples from a distribution. As such, the MAE vs. FS comparison doesn't fully address the generative vs. discriminative distinction I was referring to. I was referring to a more direct comparison with contemporary generative models like diffusion models or autoregressive image models.
>
> [1] Sorscher et al. "Neural representational geometry underlies few-shot concept learning." PNAS 2022.
> [2] Kondapaneni et al. "Representational Similarity via Interpretable Visual Concepts." ICLR 2025.

---

> > ### Author Response · Authors · 2025-08-04
> >
> > Thank you for the thoughtful follow-up and for clarifying your perspective on novelty and the generative/discriminative distinction.
> >
> > We appreciate that Sorscher et al. and Kondapaneni et al. are closely related to our work. Sorscher et al. explore manifold concept  approximated as high-dim. ellipsoids, in a concept-supervised setting, whereas our approach discovers arbitrarily shaped concepts fully unsupervised. Kondapaneni et al. focus on class-specific linear concept analyses, which nicely complement our more global, non-linear alignment analysis. While we cite Kondapaneni et al. in our Related Works, we will add the work by Sorscher et al.
> >
> > We also appreciate your clarification regarding the generative–discriminative distinction. Extending NLMCD-CBA to modern generative models such as diffusion or autoregressive models would indeed be the next step to provide the insights you highlight.
> >
> > Thank you again for the constructive feedback. We will incorporate your suggestions on emphasizing practical insights and broader comparisons in the final version.

---

### Official Review · Reviewer_V5Qu · 2025-07-03

**Clarity:** 2
**Significance:** 2
**Originality:** 3
**Rating:** 5
**Confidence:** 5

**Summary:**

- This paper proposes an interesting analysis pipeline for investigating and exploring the alignment properties of representations in Vision Transformers. Specifically, unlike existing approaches that reduce alignment to a single scalar value, the authors introduce a concept-based alignment metric that treats concepts as nonlinear manifolds. In practice, they use a soft clustering method, i.e., HDBSCAN applied on the UMAP projection, as a proxy for identifying these concept manifolds. This metric is then used to evaluate alignment across fuzzy concept clusters derived from different architectures. The proposed pipeline is applied to four ViT models, i.e., FS, CLIP, DINO, and MAE, revealing insightful structural differences. The analysis enhances the interpretability of Vision Transformers and highlights their potential in downstream tasks.

**Questions:**

- Appendix B describes the design choices for hyperparameters such as UMAP’s min_dist, n_neighbors, and embedding dimensionality, as well as HDBSCAN’s min_cluster_size and min_samples, based on maximizing the DBCV score. It would be helpful to include a hyperparameter sensitivity study to show how variations in these settings affect metrics like noise rate, DBCV, NRMSE, and CBA alignment scores. This would further clarify the robustness and stability of the proposed method.
- The proposed analysis pipeline is suitable to be extended to multimodal foundation models, or compare the image and text encoders in CLIP, and assess how their fusion affects representational alignment?

**Ethical Concerns:**

["NO or VERY MINOR ethics concerns only"]

**Final Justification:**

After reading the comments from the other reviewers and the authors' rebuttal, I find that most of my concerns have been addressed, as noted in my earlier response. Overall, the paper presents a clear and technically sound analysis pipeline that offers fine-grained insights beyond scalar alignment scores, and the rebuttal effectively resolves the key practical questions. Therefore, I will maintain my current rating.

**Limitations:**

Yes.

**Paper Formatting Concerns:**

No paper formatting concerns.

**Quality:**

2

**Strengths And Weaknesses:**

# Strengths

- The proposed method shows strong potential for future extensions.
- The approach is conceptually simple and intuitively appealing.
- The paper presents the proposed methodology in a clear and well-structured manner.

# Weaknesses
- The proposed method appears to be time-consuming, as it requires traversing all vectors. However, the authors do not provide any analysis of the computational cost, which undermines the practical value and contribution of the method.
- The paper relies on ImageNet class labels and coarse WordNet categories as weak proxies to annotate discovered concepts, which may not accurately capture the true semantic content of each concept manifold. Moreover, in the CLIP experiments, the analysis is confined to the visual encoder; extending the analysis to CLIP’s text encoder and to the joint image–text embedding space could reveal richer multimodal alignment patterns and better validate the method’s generality.

---

> ### Author Rebuttal · Authors · 2025-07-31
>
> Thank you for the positive assessment of our method as conceptually simple and intuitively appealing with strong potential for future extensions. We are excited to apply our method in other use-cases in the future, especially for the analysis of multi-modal models.
>
> **[W1] Computational cost**
>
> We agree that computational cost of computing CBA is a limitation of our method, and we will add this to the Limitations paragraph. In practice, we subsample the feature vectors and use batched matrix operations to reduce runtime. In appendix B, we report runtime on a V100 GPU (concept-wise CBA for representations with around 500 concepts takes 6 h).
>
> **[W2]  1) WordNet categories**,  2) confinement to CLIP image encoder – see [Q2]
>
> We clarify that coarse labels are used only for visualizing the concept atlas in Fig. 4. We agree that they might not capture the true semantics of the concept manifold and acknowledge their limitations in a footnote in L. 269 (we will fix the formatting issue with this footnote).  In future work, we aim to build an interactive tool to better visualize the concept atlas, where the user can click on each concept to see a visualization. At this point we also want to clarify that class label alignment in Fig. 5 is based on the more fine-grained original Imagenet labels.
>
> **[Q1] Hyperparameter sensitivity**
>
> We performed  a small sensitivity study (at layer 11 of DINO SEQ representations) that varies UMAP and HDBSCAN parameters one at a time,  and observe that our concept discovery evaluation metrics are reasonably stable within a quite broad hyperparameter range.
>
> UMAP hyperparameters and embedding/clustering quality:
>
>
>
> | Hyperparameter           | Range       | NRMSE (min < **default** < max) | Noise ratio            | DBCV                   | Robustness             |
> | ------------------------ | ----------- | ------------------------------- | ---------------------- | ---------------------- | ---------------------- |
> | **Embedding dim (zdim)** | 20 – 100    | 0.7 < **9.5** < 15.1       | 0.66 < **0.66** < 0.66 | 0.56 < **0.58** < 0.59 | 0.84 < **0.84** < 0.87 |
> | **min\_dist**            | 0.005 – 0.5 | 0.7 < **9.5** < 13.6        | 0.64 < **0.66** < 0.72 | 0.53 < **0.58** < 0.58 | 0.81 < **0.85** < 0.86 |
> | **n\_neighbors**         | 10 – 40     | 6.4< **9.5**         | 0.65 < **0.66** < 0.66 | 0.58 < **0.58** < 0.61 | 0.84 < **0.85** < 0.85 |
>
> HDBSCAN hyperparamters and clustering quality:
>
> | Hyperparameter         | Range     | Noise ratio            | DBCV                   | Robustness             |
> | ---------------------- | -------- | ---------------------- | ---------------------- | ---------------------- |
> | **min\_cluster\_size** | 20 – 100          | 0.57 < **0.66** < 0.73 | 0.57 < **0.58** < 0.59 | 0.83 < **0.84** < 0.85 |
> | **min\_samples**       | 10 – 50       | 0.66 < **0.66** < 0.67 | 0.50 < **0.58** < 0.65 | 0.83 < **0.84** < 0.85 |
>
>
> Only NRMSE shows a rather high range when varying zdim and min_dist with minimum <1.0 for small embedding dimensions (<=40) and very small min_dist. This is likely not a sign of a superior embedding, but rather an unintuitive behaviour of this metric due to extreme compression of pairwise distances in the embedding.
>
> We will add the results of this study to the appendix.
>
> **[Q2] Comparison of image and text encoder in CLIP**
>
> We see cross modal concept based alignment as an exciting application of NLMCD CBA, which is directly applicable to the CLIP text encoder. Due to the limited rebuttal period, we cannot present results here, but here is what we would do:  We would focus on CLS token representations, as image localized concepts (e.g., a cat’s whiskers) do not directly correspond to textual mentions (e.g., “cat with whiskers”). Further, we hypothesize that cross modal concept alignment is most meaningful at the final layers, where image representations are more abstract and more likely to align with the concepts described in the associated text.

---

> > ### Comment · Reviewer_V5Qu · 2025-08-05
> > **Response of the rebuttal**
> >
> > Thank you to the authors for the comprehensive rebuttal. After reviewing the response and the ensuing discussion, I am satisfied that my main concerns have been thoroughly addressed. The authors have now provided concrete runtime numbers and have committed to including these details in the Limitations section, which alleviates concerns regarding computational cost. They also present a hyperparameter sensitivity study demonstrating that key metrics remain stable across a broad range of UMAP/HDBSCAN settings, further reinforcing the method’s robustness. Clarifications regarding the use of WordNet labels stating that they are solely for visualization purposes, and the switch to fine-grained ImageNet labels in Figure 5 satisfactorily resolve the annotation issue. Although cross-modal CBA evaluation for CLIP is left for future work, the proposed plan is reasonable and does not detract from the current paper’s vision-only contributions. Overall, the paper delivers a clear and technically solid analysis pipeline that provides fine-grained insights beyond scalar alignment scores, and the rebuttal effectively resolves the key practical questions. I am therefore maintaining the rating.

---

### Official Review · Reviewer_ErWZ · 2025-07-04

**Clarity:** 3
**Significance:** 3
**Originality:** 3
**Rating:** 5
**Confidence:** 3

**Summary:**

This paper proposes a novel framework to analyze and compare deep visual representations at a conceptual level rather than with a single similarity score. The authors combine concept discovery (finding interpretable clusters or “concepts” in a model’s latent space) with representational alignment analysis, yielding fine-grained insights into how different layers or models align in terms of learned features. Key contributions include: (1) defining concepts as non-linear manifolds in feature space (as opposed to traditional linear directions), and developing a clustering-based method to identify these concept manifolds (using UMAP for dimensionality reduction and HDBSCAN for unsupervised clustering); (2) introducing a concept-based alignment (CBA) measure using a generalized Rand index that quantifies similarity between two sets of concept clusters, which can be decomposed into pairwise distances between individual concepts; and (3) applying this analysis to several Vision Transformers (ViTs) trained with different paradigms (supervised, contrastive, self-supervised, etc.), revealing how their internal representations differ. Notably, the paper finds that certain high-level concepts are universal across models, while others are model-specific, and that increased training supervision tends to reduce the semantic organization of learned features. The proposed approach thus goes “beyond scalars” like CKA by pinpointing which concepts are shared or divergent between representations, offering a more interpretable comparison of model internals.

**Questions:**

1. Semantic coherence of discovered concepts: By qualitative inspection, the discovered concept clusters (e.g. an “apples” concept in Fig. 3) appear semantically coherent, with the nonlinear (NLMCD) clusters showing high within-cluster consistency. How do the authors evaluate or ensure that these data-driven concepts are indeed meaningful and interpretable to humans?

2. Fine-grained CBA vs. scalar alignment (CKA): The proposed Concept-Based Alignment (CBA) measure achieves performance on par with standard scalar alignment measures like CKA in the sanity check, while additionally providing a fine-grained view into how individual concepts align. What new insights does this concept-level alignment enable that a single scalar (e.g. CKA) might obscure? For instance, have the authors observed cases where two models or layers have similar overall alignment scores yet differ in which specific concepts are aligned (or misaligned)? Highlighting such a scenario would clarify the practical advantage of CBA’s fine-grained analysis over traditional one-number similarity metrics.

**Ethical Concerns:**

["NO or VERY MINOR ethics concerns only"]

**Final Justification:**

I thank the authors for their detailed rebuttal. Many of my concerns have been adequately addressed. The additional discussion comparing concept-based alignment and scalar-based alignment was particularly insightful. Overall, I find the paper to be of sufficient quality for acceptance, and I have decided to maintain my original score.

Additionally, I encourage the authors to consider the following suggestions for future work, which may further enhance the impact and generality of their approach:

1. Extend the experiments to a broader range of model architectures and language models to better demonstrate the generalizability of the proposed method.

2. Investigate potential failure cases where the scalar alignment score does not accurately reflect model behavior, but the concept-based alignment does. Such analysis would further highlight the advantages and robustness of your approach.

**Limitations:**

See weakness and questions

**Quality:**

3

**Strengths And Weaknesses:**

Strengths:
1. Innovative Fine-Grained Analysis: The paper addresses a clear limitation of existing representational similarity measures (e.g. CKA) which collapse complex relationships into a single number. By integrating concept discovery with alignment, the method yields rich, interpretable information about what is similar between layers or models (i.e. which concepts align). This is a novel perspective that provides insights not available through traditional scalar metrics.
2. Novel Concept Definition: Treating concepts as non-linear manifolds in feature space is an innovative generalization beyond prior linear subspace definitions. This more flexible definition is well-motivated (aligning with the manifold hypothesis) and the authors demonstrate its advantage: in a controlled “sanity check,” their non-linear concept-based alignment outperforms simpler linear concept approaches in capturing expected layer-wise similarity patterns.
3. Insightful Findings for Vision Transformers: Using the proposed analysis, the paper provides interesting observations about ViT models. For instance, self-supervised models (DINO, MAE, CLIP) develop strongly semantic, coherent concept structures in later layers, whereas the fully supervised ViT (FS) shows less semantic organization in its top-layer concepts.
4. Clarity and Organization: The paper is well-structured and easy to follow.

Weakness:
1. Complexity and Scalability: A potential concern is the computational complexity of the concept discovery pipeline. The approach relies on UMAP for embedding and HDBSCAN for clustering, which required the authors to subsample the dataset (25% of ImageNet and spatial token pooling) due to scalability limits. This subsampling raises the question of whether some fine concepts might be missed or labeled as “noise” by HDBSCAN when data is limited.
2. Scope Limited to ViTs: The experiments focus on four ViT models with a specific architecture and on a single dataset (ImageNet). It remains unclear how well the proposed alignment analysis generalizes to other network architectures (e.g., CNNs or different ViT variants) or other data domains.

---

> ### Author Rebuttal · Authors · 2025-07-31
>
> Thank you for your helpful feedback and describing our analysis as innovative, novel, and insightful.
>
> **[W1] Computational Complexity**
>
> We agree that the UMAP+HDBSCAN pipeline is compute-intensive and list this as a limitation. In future work, we want to explore more scalable alternatives, such as deep density-based clustering (e.g., Beer et al., 2024), which may allow efficient clustering at scale.
>
> **[W2] Scope**
>
> While our experiments focus on ViTs, we note that our method is model-agnostic and applicable to other architectures. We plan to extend to CNNs and and in particular to language models in future work.
>
> **[Q1] Concept interpretability**
>
> NLMCD is not optimized for human interpretability per se — it captures concept manifolds that are meaningful from the structure of the model's representation. To assess human interpretability of the discovered concepts, we inspected the concepts ourselves. In the future, it would be interesting to evaluate this in more depth, by performing a human user study in the spirit of Ghorbani et al. (2019) or use a foundation model as a proxy human like Dreyer et al. (2025).
>
> **[Q2] Insights from concept-based alignment**
>
> Thanks for this helpful prompt to further demonstrate the additional insights that fine-grained concept alignment offers. Based on your question about concept-level differences between pairs of representations that share the same overall alignment score, we performed the following analysis:
>
> Layer 10 in MAE/DINO (see Fig. 6) and the same layer in CLIP/DINO (see Fig. 16) share the same overall alignment (0.74). To summarize fine-grained differences in concept-alignment, we grouped concept nearest-neighbor-pairs into the WordNet categories from Fig. 4 (88% of concept pairs are in the same category for MAE/DINO, 93% for CLIP/DINO). We then computed the mean concept distance within each concept category and find that 1) canine and insect concepts have high distance in both model pairs, while equipment and material concept have low distance in both pairs; 2) fish concepts are similar in MAE/DINO but have high distance in CLIP/DINO, while sport concepts show the opposite trend.
>
> This exemplifies a scenario where two representations are similarly aligned by the overall score, but fine-grained concept alignment differs. We will add this analysis to the appendix.
>
> Beer et al., 2024: A. Beer et al., "SHADE: Deep Density-based Clustering," 2024 IEEE International Conference on Data Mining (ICDM), Abu Dhabi, United Arab Emirates, 2024, pp. 675-680
>
> Ghorbani et al. 2019: Ghorbani, A., Wexler, J., Zou, J.Y., & Kim, B. (2019). Towards Automatic Concept-based Explanations. Neural Information Processing Systems.
>
> Dreyer et al. 2025: Dreyer, M., Berend, J., Labarta, T., Vielhaben, J., Wiegand, T., Lapuschkin, S., & Samek, W. (2025). Mechanistic understanding

---

> > ### Comment · Reviewer_ErWZ · 2025-08-05
> > **Reply to the authors**
> >
> > I thank the authors for their detailed rebuttal. Many of my concerns have been adequately addressed. The additional discussion comparing concept-based alignment and scalar-based alignment was particularly insightful. Overall, I find the paper to be of sufficient quality for acceptance, and I have decided to maintain my original score.
> >
> > Additionally, I encourage the authors to consider the following suggestions for future work, which may further enhance the impact and generality of their approach:
> >
> > 1. Extend the experiments to a broader range of model architectures and language models to better demonstrate the generalizability of the proposed method.
> >
> > 2. Investigate potential failure cases where the scalar alignment score does not accurately reflect model behavior, but the concept-based alignment does. Such analysis would further highlight the advantages and robustness of your approach.

---

### Official Review · Reviewer_8f6e · 2025-07-12

**Clarity:** 2
**Significance:** 3
**Originality:** 4
**Rating:** 5
**Confidence:** 4

**Summary:**

The authors propose a new method of concept level alignment analysis in pre-trained vision transformers. The first part of the author's solution involves building a manifold based approach of concept discovery in neural network representations which they call NLMCD: first they reduce the representation dimensions using UMAP, and then perform heirarchical clustering using HDBSCAN. This leads to 'soft' assignment of feature representations to different concept clusters. In order to develop an alignment metric using their soft clustering, the authors adapt a generalization of Rand index - the distance between two concept clusters is given as an average of distances between two membership vectors. The authors define concept-based alignment (CBA) as difference of this averaged distance from 1.The authors perform several quantitative checks on their concept discovery approach, and the results from these are somewhat mixed. A sanity check on the CBA metric is performed against CKA which shows that the author's metric offers more fine-grained alignment scores. Lastly, and perhaps the most impactful section of the work, the authors perform concept level alignment analyses on four different pre-trained models on ImageNet and confirm several results from representation level analyses in the past.

**Questions:**

1. In Line 187 "we test the robustness of our approach by measuring the alignment between two runs (with different initializations)" can the authors confirm what they mean by different initializations - are these different initializations of the clustering, or of the vision transformer training, or something else?

2. It is unclear to me why the authors consider the SEQ tokens at all, and especially after pooling across layers, which would lead to signficant information loss in terms of granularity of concepts learnt. Can the authors explain why they examined SEQ tokens?

3. In the comparison of vision models, please include more references beyond just analysis of attention patterns, CKA, and downstream performance. A lot of research has been done on comparing vision model representations as well as concepts across different lenses (e.g. training objectives, dataset sizes, downstream tasks, transfer learning, multi-modal learning).  Some of these lines of research (not exhaustive) are in these papers:

* On the duality between contrastive and non-contrastive self-supervised learning, Garrido et al
* Objectives Matter: Understanding the Impact of Self-Supervised Objectives on ViT Representations, Shekhar et al
* A large-scale examination of inductive biases shaping high-level visual representation in brains and machines, Conwell et al
* Objective drives the consistency of representational similarity across datasets, Ciernik et al

4. How are the concept formation graphs (CFGs) made in Fig 3? As in how did the authors figure out the layer 7-9 concept development - was this through manual observation, or was there some quantitative indicator e.g.

**Ethical Concerns:**

["NO or VERY MINOR ethics concerns only"]

**Final Justification:**

The authors have provided adequate responses to the questions raised in my review. I think the main merit of the paper is in the problem its trying to address, which is both important and underexplored. The method used is quite novel, and while not scalable, the results are important enough to drive further research into concept based alignment.

**Limitations:**

yes

**Paper Formatting Concerns:**

For Figure 2, it is hard to make out useful information due to several small panels as well as the overlap of SEQ and CLS based plots on the same images. If possible, please try to break down the images into more panels or images, and reduce the amount of information presented in each panel.

**Quality:**

3

**Strengths And Weaknesses:**

**Strengths**

1. Concept-based alignment instead of representation-based alignment is a better approach towards quantifying representation similarities across neural networks, training objectives, and training distributions. The authors approach is among one of the first methods towards this, and as such quite novel.
2. The concept alignment analysis is quite interesting, the authors' analysis helps to re-establish some representation level results for feature learn through concept level analysis too (e.g. inter-modal concept alignment in early layers, similarity in concepts across similar training objectives.)

**Weaknesses**

1. The algorithm used for calculating concept discovery (UMAP followed by HDBSCAN) is very compute intensive, which is reflected in the author's experimental choices - using 25% of ImageNet, pooling sequence tokens, perhaps the use of UMAP itself. It will be hard to translate this concept discovery method (and hence alignment metric) to web scale datasets, where alignment is perhaps more important as data quality is not controlled.
2. The experimental results for concept discovery evaluation are not presented well, and also have a lot of variance across models. Why does the NRMSE drop for intermediate layers for some models? Yet other models like CLIP have their values same across all layers? Robustness decreasing with layers is generally not a positive sign, although the absolute values are themself high enough where it might not be an issue. Overall, it is not clear to me as a reader what inference and conclusions to draw from the author's results. Perhaps the section 4.1 evaluation can be broken down into evaluation and results separately to explain the results better.

---

> ### Author Rebuttal · Authors · 2025-07-31
>
> We appreciate the assessment of our method as novel and our concept alignment analysis as interesting, and thank you for the helpful feedback.
>
> **[W1] Computational Complexity**
>
> We agree that the UMAP+HDBSCAN pipeline is compute-intensive and list this as a limitation. In future work, we want to explore more scalable alternatives, such as deep density-based clustering (e.g., Beer et al., 2024), which may allow efficient clustering at scale.
>
> **[W2] Presentation of concept discovery evaluation**
>
> We appreciate the suggestion to clarify this section and will improve separation between evaluation and results.
> Regarding the drop in DBCV for the FS model around layer 9 (and the smaller drop at layer 6 in DINO), we note that these layers coincide with a pronounced increase in the number of discovered concepts (Fig. 5). In contrast, as you notice, the increase of concepts across layers is smoother for CLIP, and also MAE.
> We hypothesize that this reflects a sudden reorganization of concepts, which makes the clustering problem at these layers  more challenging before the representation exhibits a new structure.
>
> **[Q1] Evaluation of Robustness**
>
> We tested robustness by re-running the clustering pipeline with different random seeds (i.e., different UMAP/HDBSCAN initializations) while keeping model weights and input samples fixed. We will clarify this in Line 187.
>
> **[Q2] SEQ token**
>
> We analyze spatial (SEQ) tokens to localize emerging concepts in the input space. ViT patch tokens are widely observed to retain spatial correspondence (e.g., Caron et al. 2021), so clustering SEQ tokens allows us to capture localized concepts. We pool only across spatial tokens to obtain a representative sample across all ImageNet images, and never across layers.
>
> **[Q3] Related work**
>
> Thank you for the suggestions, we will add these papers to our Related Works section.
>
> **[Q4] CFGs**
>
> The textual concept description in Fig. 3 are maually crafted, but the graph itself is constructed via in algorithm described in appendix D. In short, we compute transition rates of tokens between concept clusters at two consective layers, an then construct a binary unidirectional graph with edges between those concept nodes for which the respective transition rate surpasses a thresold.
>
> Beer et al., 2024: A. Beer et al., "SHADE: Deep Density-based Clustering," 2024 IEEE International Conference on Data Mining (ICDM), Abu Dhabi, United Arab Emirates, 2024, pp. 675-680
>
> Caron et al. 2021: Mathilde Caron, Hugo Touvron, Ishan Misra, Hervé Jégou, Julien Mairal, Piotr Bojanowski, and Armand
> Joulin. Emerging properties in self-supervised vision transformers. In International Conference on Com-
> puter Vision, pp. 9630–9640, 2021

---

> > ### Comment · Reviewer_8f6e · 2025-08-06
> >
> > Thank you for the rebuttal. Most of my concerns around positioning and issues with experimental results have been addressed. I will keep my original judgement - I think this paper should be accepted and would make a good addition to the fields of semantic concept discovery and alignment research.

---

### Decision · Program_Chairs · 2025-09-17

**Decision:**

Accept (poster)

**Comment:**

This submission proposes to use UMAP followed by HDBSCAN clustering to cluster activations of neural networks, and then to use the similarity between the clusterings of examples across neural networks to measure representational alignment. The resulting clusterings are interpretable and seems sane insofar as it assigns higher similarity to nearby layers than to distant ones. When applied to four ViT models trained with different forms of (self-)supervision, the proposed method reveals that earlier layers are more similar across models than later ones, and demonstrating that the method allows to decompose the scalar similarity measure into similarities between pairs of concepts.

Reviewers felt that the approach to representational similarity was novel and promising, and the comparison of ViTs was interesting. They noted that the method is very computationally intensive, but did not feel that this limitation was sufficient to undermine the utility of the method.

I agree with the reviewers that the proposed method seems promising and the evaluation is convincing insofar as the general sanity of the method. It would be nice if the submission offered a clearer demonstration that the method can be used to discover new insights into the differences between models that can then be independently validated by some other means. But overall, this is a valuable contribution that introduces a novel and interpretable approach to measuring representational alignment, with clear potential for future applications.